# Optimization and Bayes: A Trade-off for Overparameterized Neural Networks

**Zhengmian Hu, Heng Huang**
Department of Computer Science
University of Maryland
College Park, MD 20740
huzhengmian@gmail.com,henghuanghh@gmail.com

## Abstract

This paper proposes a novel algorithm, Transformative Bayesian Learning (TransBL), which bridges the gap between empirical risk minimization (ERM) and Bayesian learning for neural networks. We compare ERM, which uses gradient descent to optimize, and Bayesian learning with importance sampling for their generalization and computational complexity. We derive the first algorithm-dependent PAC-Bayesian generalization bound for infinitely wide networks based on an exact KL divergence between the trained posterior distribution obtained by infinitesimal step size gradient descent and a Gaussian prior. Moreover, we show how to transform gradient-based optimization into importance sampling by incorporating a weight. While Bayesian learning has better generalization, it suffers from low sampling efficiency. Optimization methods, on the other hand, have good sampling efficiency but poor generalization. Our proposed algorithm TransBL enables a trade-off between generalization and sampling efficiency.

## 1 Introduction

Deep Neural Networks (DNNs) have achieved remarkable success in machine learning and related applications. It repeatedly outperformed conventional machine learning approaches, resulting in ground-breaking research such as human-level performance in computer vision [31], substantial progress in natural language processing [10], and mastering the game of Go [65].

The success of DNNs is the result of a critical combination of the complexity and generalization. On the one hand, universal approximation theorem [34] guarantees that any continuous function can be approximated arbitrarily well by using a deep network. On the other hand, deep architectures, together with large-scale training data and back-propagation algorithm, present a good generalization ability towards unseen data.

Although expressive power and generalization ability are both desirable on their own, they are mutually incompatible. Being one of the main contributions of statistical learning theory, the probably approximately correct (PAC) learning [69] allows us to establish an upper bound of generalization gap by capacity of a hypothesis space. Generally speaking, if a model is capable of fitting random labels, then it must generalize poorly.

Recent result [79] shows that overparameterized DNNs do fit random label perfectly. However, it is observed that more steps of stochastic gradient descent (SGD) are needed to train a neural network to fit random labels. This phenomenon suggests that the learning capability of DNNs increases with the number of training steps. In light of this observation, the algorithm-dependent generalization bounds which control the complexity by maximum training steps are preferred over uniform convergence bounds.

37th Conference on Neural Information Processing Systems (NeurIPS 2023).

In this paper, we follow the PAC-Bayesian approach to derive our algorithm-dependent generalization bounds. PAC-Bayesian bounds were first introduced by McAllester [53] and further developed in [61, 52, 13]. These bounds apply for stochastic learning algorithms and focus on expected generalization error over a probability on parameter space. The generalization is related to the KL divergence between the output distribution of a learning algorithm and a prior distribution.

Typically, deep neural networks contain a large number of parameters and are trained by numerous training epochs through some gradient descent updates. The training dynamics of deep neural networks is complicated, and hence direct theoretical analysis on the KL divergence is intractable. Fortunately, such situation can be largely simplified in the infinite width limit [55, 73, 42, 51, 18, 39, 43, 76, 3]. Under this limit, the output distribution of stochastic neural networks drawn from the prior is approximated by a Gaussian distribution. Moreover, the training dynamics of gradient-based optimization is governed by the kernel gradient descent, which guarantees that the evolution of network output only depends on the function values themselves. Thus, our first problem to study in this paper is: **Q1 –** *In the infinitely wide neural network limit, can we theoretically derive the formula of this KL divergence?*

Optimizing the PAC-Bayesian bound of expected error gives rise to the Gibbs measure. It is also widely termed as posterior, as it shares the same form as posterior in Bayes' rule, given that error is interpreted as likelihood. Drawing samples from this posterior is very hard in practice. Markov Chain Monte Carlo (MCMC) methods have been explored for deep Bayesian learning. However, it suffers from slow convergence and high computational cost on high- dimensional parameter spaces. Non-exact minimization of PAC-Bayesian bound gives rise to variational approximation (VA) which is more computationally efficient, but biased due to the difference between variational distribution and true posterior.

Given that MCMC and VA have their own disadvantages in Bayesian learning, our paper investigates the second problem: **Q2 –** *Does there exist a Bayesian learning method with non-diminishing sampling efficiency even for infinite wide neural network?*

Finally, we notice that the gradient-based optimization is efficient in training DNNs, but comes with larger generalization error. Bayesian learning, on the other hand, optimizes the expected loss but has lower efficiency. Our third question to study is: **Q3 –** *Does there exist an interpolation between optimization and Bayesian learning for trade-off between computation efficiency and generalization error?*

In this paper, we give positive answers to above three questions and the main contributions of this paper are summarized as follows:

1) We analyze the infinitely wide neural network trained by gradient flow, or equivalently infinitesimal step size gradient descent. We show that the infinite width limit largely simplifies the training dynamics, and the KL divergence between the output distribution and Gaussian prior can be formulated as a function of training time and training data.

2) As a byproduct of our analysis on the generalization and sampling efficiency, we prove that the trace of Hessian for DNN is not diminishing under infinite width limit and depends on the initialization and training. To the best of our knowledge, this dynamics of Hessian trace is new and maybe of independent interest.

3) We show that if the determinant of Jacobian of optimization flow is available, we can compute a weight for each optimized predictor, such that the weighted output distribution is just posterior. The sampling efficiency in infinite width limit is also derived. We call this type of algorithm as Transformative Bayesian Learning (TransBL) because it is transformed from an optimization procedure.

4) We show that modifying the additional weight in TransBL gives rise to an interpolation between optimization and Bayesian learning. The behaviour of TransBL is increasingly similar to optimization when the weight is changed toward being uniform. This interpolation doesn't alter training dynamics, thus enables flexible trade-off between sampling efficiency and generalization.

## 2  Background

We first explain the setting and briefly review PAC-Bayesian and Bayesian learning as background. Other related works is discussed in Appendix A.

**Problem Setup**   Given input space $\mathcal{X}$ and label space $\mathcal{Y}$, we assume that labeled training data are drawn independently from an unknown data distribution $D$ over $\mathcal{X} \times \mathcal{Y}$. The training set is denoted as $S^{\text{train}} = \{(s_a, z_a)|1, \ldots, m\}$, where $m$ is the size of the training sample. A predictor is a function $h : \mathcal{X} \to \mathbb{R}$ and the set of predictors is denoted as $\mathcal{H}$. We introduce two losses $l : \mathbb{R} \times \mathcal{Y} \to [0, 1]$ and $l^s : \mathbb{R} \times \mathcal{Y} \to \mathbb{R}$, where the first one is used to measure the error rate and the second smooth surrogate loss with polynomially bounded second order derivative is used for providing learning signal in gradient-based training. We define expected loss as $R(h) = \mathbb{E}_{(s,z)\sim D}[l(h(s), z)]$ and empirical loss as $r(h) = \frac{1}{m}\sum_{a=1}^m l(h(s_a), z_a)$. The expected loss is the central quantity which we are interested in but cannot be unobserved in general. Therefore, the currently popular approach to statistical learning is to derive an upper bound of expected loss.

**PAC-Bayesian Bounds**   We will follow the PAC-Bayesian approach to the generalization problem.

**Theorem 1** (Theorem 1.2.6. in [13])**.** *For any positive $\lambda$ and $\delta \in (0, 1)$, with at least $1 - \delta$ in the probability of training samples, for all distribution $q$ on space $\mathcal{H}$, we have*

$$\mathbb{E}_{h \sim q}[R(h)] \leq \Phi_{\frac{\lambda}{m}}^{-1}\left(\mathbb{E}_{h \sim q}[r(h)] + \frac{D_{\text{KL}}(q\|p) + \log\frac{1}{\delta}}{\lambda}\right).$$

*The KL divergence is $D_{\text{KL}}(q\|p) = \mathbb{E}_{x \sim q}[\log\frac{q(x)}{p(x)}]$ and the auxiliary function is defined as*

$$\Phi_a^{-1}(v) = \frac{1 - \exp(-av)}{1 - \exp(-a)} \leq \frac{av}{1 - \exp(-a)}.$$

Although there are huge amounts of research on PAC-Bayesian approach, all of them rely on the KL divergence between output distribution $q$ of stochastic predictor learned from certain algorithm and a prior $p$, therefore the analysis of this KL divergence is a natural question.

**Bayesian Learning**   We define Gibbs measure as $p_\lambda(\theta) = \frac{1}{Z_\lambda}e^{-\lambda r(\theta)}p(\theta)$ and partition function as $Z_\lambda = \mathbb{E}_{\theta \sim p}[e^{-\lambda r(\theta)}]$. The Gibbs measure is related to posterior in Bayesian inference [24], and minimize right-hand side in theorem 1, inducing a much tighter bound:

$$\mathbb{E}_{h \sim q}[R(h)] \leq \Phi_{\frac{\lambda}{m}}^{-1}(-\frac{1}{\lambda}\log(Z_\lambda\delta)) \leq \frac{1}{m(1 - \exp\left(-\frac{\lambda}{m}\right))}\log\frac{1}{Z_\lambda\delta}.$$

Despite the fact that expected loss is optimized, Bayesian learning bears significantly more difficulty than popular optimization-based algorithms. It is noted that the sampling from a high dimensional distribution with irregular shape may lead to high fluctuations and low efficiency, yet in typical neural network, the number of parameters to be trained is very large, and loss landscape is complicated. Many sampling techniques have been developed and explored, but the state-of-the-art is still far from satisfactory. In this paper, we concentrate on theoretical understanding of the Bayesian approach and consider a special class of importance sampling in Section 4 where we can compare the computation efficiency of Bayesian method and optimization approach.

## 3   Training by Optimization

We now derive the KL divergence under deterministic optimization setting. We assume all predictors are parameterized by some parameters $\theta \in \Theta$. The expected loss $R(h)$ and empirical loss $r(h)$ could therefore be regarded as functions of parameter $\theta$. An optimization flow is a function $f : \Theta \to \Theta$, which typically relies on the training data to update the model's parameters $\theta$ and minimize the empirical loss $r(h)$. The updating can be achieved through various methods such as single or multiple steps of gradient descent, gradient flow with infinitesimally small step sizes, or more sophisticated methods detailed in Appendix B. We require the Jacobian determinant $J_f(\theta) = \det(\nabla f(\theta))$ exists and the optimization flow is a bijective, *i.e.* the inverse $f^{-1}$ exists everywhere. This is satisfied for gradient flow and gradient descent when step size is smaller than $\frac{1}{L}$. More situations where this assumption holds is discussed in Appendix B. The starting point of optimization is initialized from a prior distribution $p(\theta)$, and we define a corresponding energy $V(\theta) = -\log p(\theta) + C$ where $C$ is a parameter independent constant. The output distribution $q$ could be characterized as follows:

$$\Pr_q(\theta \in A) = \Pr_p(f(\theta) \in A), q(\theta) = p(f^{-1}(\theta))\big|J_{f^{-1}}(\theta)\big|.$$

The KL divergence between output distribution and prior can be derived as follows:

$$D_{\text{KL}}(q\|p) = \mathbb{E}_{\theta\sim q}[\log\frac{q(\theta)}{p(\theta)}] = \mathbb{E}_{\theta\sim p}[\log\frac{q(f(\theta))}{p(f(\theta))}] = \mathbb{E}_{\theta\sim p}[\log\frac{p(\theta)}{p(f(\theta))} - \log|J_f(\theta)|].$$

Based on a physics analog of Helmholtz free energy change in isothermal process, we define energy term and entropy term separately:

$$\Delta_f V(\theta) = V(f(\theta)) - V(\theta), \quad \Delta_f S(\theta) = \log|J_f(\theta)|.$$

The energy term only depends on the prior. If Gaussian prior is used, $\Delta_f V(\theta)$ corresponds to the change of squared norm of parameters. Although the KL divergence is positively related to the energy term, and energy term is monotonically increasing with norm of trained parameters, naively compressing parameters to reduce parameter norm doesn't lead to decrease in KL divergence, because the decrease in energy term is offset by the change in entropy term.

The entropy term determines the gain or loss of entropy caused by an optimization procedure. The pure gain means that the optimization flow maps a region of parameter space into another region with larger area. For example, consider the simple function $f(x) = 2x$ in a one-dimensional setting. The Jacobian of this function, which describes how much the function stretches or compresses space, is constant at $2$. Imagine we start with an input $x$ that is uniformly distributed within the interval $[0, 1]$. After applying the function, the output $f(x)$ becomes uniformly distributed over this larger interval $[0, 2]$. Therefore the entropy increase by $\log(2)$. However, most optimization procedures pursue minimization of certain objective function which is only small in an extremely small region of parameter space, and thus loss of entropy is inevitable. In Section 7, we show that for gradient descent algorithm, the loss of entropy is related to the local curvature. We also extend the discussion to algorithms with enriched state space, such as Momentum SGD and Adagrad in Appendix F.2.

Finally, we have

$$D_{\text{KL}}(q\|p) = \mathbb{E}_{\theta\sim p}[\Delta_f V(\theta) - \Delta_f S(\theta)] \tag{1}$$

The above KL divergence represents an increase in free energy, or the information gain from the training, depending on the point of view. Both energy term and entropy term can be evaluated empirically, though the computational cost depends on the choice of optimization method, network architecture and prior.

## 4 Transformative Bayesian Learning

In this section, we introduce a class of importance sampling algorithms solving the Bayesian learning problem. These algorithms are based on the optimization flow, energy change, and loss of entropy discussed in Section 3.

The most simple example of TransBL algorithm comes from using the output distribution $q$ of an optimization flow $f$ as the proposal distribution. For any function $F$ on parameter space $\Theta$, we have

$$
\begin{aligned}
\mathbb{E}_{\theta\sim p_\lambda}[F(\theta)] =& \mathbb{E}_{\theta\sim q}\left[F(\theta)\frac{p_\lambda(\theta)}{q(\theta)}\right] = \frac{1}{Z_\lambda}\mathbb{E}_{\theta\sim p}\left[F(f(\theta))e^{-\lambda r(f(\theta))}\frac{p(f(\theta))}{q(f(\theta))}\right] \\
=& \frac{1}{Z_\lambda}\mathbb{E}_{\theta\sim p}\left[F(f(\theta))e^{-\lambda r(f(\theta))}e^{-\Delta_f V(\theta)}e^{\Delta_f S(\theta)}\right]
\end{aligned}
\tag{2}
$$

The above expectation could be regarded as attaching an additional weight to the stochastic predictor obtained from ordinary optimization, and we should use a weighted average of the results obtained by different initialization. This process doesn't involve any more training of parameter $\theta$ than the optimization procedure that TransBL is based on, though additional computation might be required for the value of $\Delta_f V(\theta)$ and $\Delta_f S(\theta)$.

The unnormalized weight of above importance sampling is $w_\lambda(\theta) = e^{-\lambda r(f(\theta))}e^{-\Delta_f V(\theta)}e^{\Delta_f S(\theta)}$. That means not all results obtained by training are treated equally. In particular, the solution with lower empirical loss, lower energy increase, and producing higher entropy is more important than others.

One benefit of TransBL is that we can measure the computation efficiency by comparing it to the training by optimization. A simple way to do this is to calculate the ratio of effective sample size and

sample size:

$$\mathrm{eff}_\lambda = \frac{(\mathbb{E}_{\theta\sim p}[w_\lambda(\theta)])^2}{\mathbb{E}_{\theta\sim p}[w_\lambda^2(\theta)]} \tag{3}$$

We define the above value as sampling efficiency, and a non-diminishing efficiency at certain limit indicates that Bayesian learning is at most a constant factor slower than optimization. The numerator of the above expression is determined by the partition function because $\mathbb{E}_{\theta\sim p}[w_\lambda(\theta)] = Z_\lambda$. Therefore, only the denominator depends on the optimization procedure, and we expect $\mathbb{E}_{\theta\sim p}[w_\lambda^2(\theta)]$ to be as small as possible for efficient Bayesian learning. Notice that this quantity again relies on energy change $\Delta_f V(\theta)$ and entropy change $\Delta_f S(\theta)$. We will give theoretical analysis of these two values for infinitely wide neural network in Section 7.

### 4.1 An Illustrative Example

For illustration purpose, we consider a univariate loss function that presents two distinct global minima with zero loss. One of these is characterized as a sharp minimum, while the other represents a flat minimum. A direct initialization from the prior, followed by training using gradient flow, often results in an ensemble with significant deviation from the posterior. This is because the optimization process fails to recognize the presence of the sharp minimum, while insights from PAC Bayesian indicate that the flat minimum is surrounded by a higher posterior probability density. TransBL method applies a small weight to the solution found within the sharp minimum. Consequently, TransBL can adeptly recreate the posterior, as shown in the Figure 6b. Due to space limitations, all figures for the illustration are moved to Appendix F.1.

## 5 Connections between Bayesian Learning and Optimization

### 5.1 A Bayesian Perspective for Optimization

We show that, in order to achieve low expected loss, the optimization flow in deep learning should reshape the initial distribution to a good estimation of posterior. A formal argument relies on applying Donsker and Varadhan's variational formula [17] to obtain the following equation:

$$\mathbb{E}_{\theta\sim q}[r(\theta)] + \frac{1}{\lambda}D_{\mathrm{KL}}(q\|p) = -\frac{1}{\lambda}\log\mathbb{E}_{\theta\sim p}[\exp(-\lambda r(\theta))] + \frac{1}{\lambda}D_{\mathrm{KL}}(q\|p_\lambda) \tag{4}$$

The first term is independent of training procedure and only relies on the definition of prior. The second term measures the KL divergence between output distribution and Gibbs measure. This is also the gap between expected loss bound obtained by some training methods and the optimal one. Notice that the above KL divergence could also be expressed in terms of weight for TransBL, energy change, and entropy loss:

$$D_{\mathrm{KL}}(q\|p_\lambda) = \log Z_\lambda + \mathbb{E}_{\theta\sim p}[-\log w_\lambda(\theta)] = \log Z_\lambda + \mathbb{E}_{\theta\sim p}[\lambda r(f(\theta)) + \Delta_f V(\theta) - \Delta_f S(\theta)]. \tag{5}$$

### 5.2 Efficiency of Optimization Flows for TransBL

In this section, we show that, in order to achieve high sampling efficiency, the optimization flow in deep learning is again expected to reshape the initial distribution to a good estimation of posterior. The only difference to the previous section is that the deviation between output distribution and posterior is measured under a different divergence.

In order to see this, we define $\chi^2$ divergence as $D_{\chi^2}(p\|q) = \mathbb{E}_{x\sim q}[(\frac{p(x)}{q(x)})^2] - 1$.

**Theorem 2.** *For sampling efficiency of TransBL whose proposal distribution is q, we have:*

$$D_{\mathrm{KL}}(p_\lambda\|q) \le \log\big(1 + D_{\chi^2}(p_\lambda\|q)\big) = -\log\mathrm{eff}_\lambda \tag{6}$$

With the above result, we can establish the equivalence between optimal generalization bound and optimal sampling efficiency, *i.e.* an optimization flow is optimal in the sense it minimizes the expected loss upper bound if and only if the sampling efficiency of TransBL is 1. A more general correspondence between generalization and sampling efficiency is desired. However, the complication lies in different divergences used in two quantities. By comparing Eq. (4) and Eq. (6),

we see that $\chi^2$ divergence is used for measuring sampling efficiency. Although an upper bound of KL divergence $D_{\mathrm{KL}}(p_\lambda\|q)$ could be obtained, this is not directly comparable to the $D_{\mathrm{KL}}(q\|p_\lambda)$ used in generalization bound because of asymmetry of KL divergence. That difference also justifies the interpolation between optimization and Bayesian learning shown in the next section.

## 6 Interpolation of Optimization and Bayesian Learning

We recall that the output distribution $q$ for the optimization and the posterior $p_\lambda$ for Bayesian learning are connected by an additional weight $w_\lambda$: $p_\lambda(\theta) = \frac{1}{Z_\lambda}w_\lambda(f^{-1}(\theta))q(\theta)$. These two distributions have distinct properties. The distribution $q$ bears worse generalization, but is easy to sample from given an established optimization oracle. The posterior $p_\lambda$, on the other hand, is better shaped, but hard to sample from. Therefore, a natural question is whether we can find a distribution $p_\lambda^\beta$ as intermediate and interpolation between them.

Given our formulation of TransBL, it turned out to be quite easy to construct such an interpolation by modifying the weight. For a function $v_\beta : \mathbb{R}^+ \to \mathbb{R}^+$ where $\beta$ is the parameter used for interpolation, we can define a modified weight as $v_\beta(w_\lambda(\theta))$. The interpolation distribution is

$$p_\lambda^\beta(\theta) = \frac{1}{Z_\lambda^\beta}v_\beta(w_\lambda(f^{-1}(\theta)))q(\theta), \quad Z_\lambda^\beta = \mathbb{E}_{\theta\sim p}[v_\beta(w_\lambda(\theta))].$$

Note that the superscript doesn't mean power but means the interpolation is controlled by $\beta$. The expectation of a function $F(\theta)$ on $p_\lambda^\beta$ could be computed similarly as Eq. (2):

$$\mathbb{E}_{\theta\sim p_\lambda^\beta}[F(\theta)] = \frac{1}{Z_\lambda^\beta}\mathbb{E}_{\theta\sim p}[F(f(\theta))v_\beta(w_\lambda(\theta))].$$

We note Eq. (4) still holds when we change $q$ into $p_\lambda^\beta$, and the gap between the expected loss bound obtained by $p_\lambda^\beta$ and the posterior is:

$$D_{\mathrm{KL}}(p_\lambda^\beta\|p_\lambda) = \log\frac{Z_\lambda}{Z_\lambda^\beta} + \frac{1}{Z_\lambda^\beta}\mathbb{E}_{\theta\sim p}\left[\log\left(\frac{v_\beta(w_\lambda(\theta))}{w_\lambda(\theta)}\right)v_\beta(w_\lambda(\theta))\right].$$

Notice that the above identity degenerates into Eq. (5) if we use $v_\beta(\cdot) = 1$, which corresponds to the original optimization.

The sampling efficiency of $p_\lambda^\beta$ is again defined as the ratio of effective sample size and sample size: $\mathrm{eff}_\lambda^\beta = (\mathbb{E}_{\theta\sim p}[v_\beta(w_\lambda(\theta))])^2/\mathbb{E}_{\theta\sim p}[v_\beta(w_\lambda(\theta))^2]$ and enjoys a relation similar to Theorem 2:

$$D_{\mathrm{KL}}(p_\lambda^\beta\|q) \le \log\left(1 + D_{\chi^2}(p_\lambda^\beta\|q)\right) = -\log\mathrm{eff}_\lambda^\beta.$$

### 6.1 Trade-off between Generalization and Sampling Efficiency

According to Eq. (5), we can see that the generalization is highly affected by extremely small $w_\lambda(\theta)$. A small weight indicates that the optimization produces certain parameters too often. This issue could be addressed by assigning a small weight for these parameters in importance sampling. On the other hand, the sampling efficiency of Bayesian learning is very sensitive to large value of $w_\lambda(\theta)$, according to Eq. (3). The high weight appears on the regime where output probability is low but posterior density is high. That essentially means the optimization oracle is not effective in exploring certain region, and the only way to bypass this bottleneck of Bayesian learning is changing the optimization oracle to produce a distribution with heavier tail than posterior. Due to the exponential dependence on energy change in weight, the distribution of weight is typically heavy-tailed, therefore the efficiency is low. Fortunately, this issue could be largely alleviated with interpolation. Since generalization bound is insensitive to large value of weight, it is possible to apply appropriate modification to weights such that the sampling efficiency is improved and generalization doesn't deteriorate too much.

Searching for the Pareto optimality among various trade-off methods is not a concern in this paper, although obviously this is an important issue and is a subject for future work. In this paper, we only consider a simple weight clipping method: $v_\beta(w) = \begin{cases} w & w \le \beta \\ \beta & \text{otherwise} \end{cases}.$

At the limit of $\beta \to 0$, $p_\lambda^\beta$ degenerates to output distribution $q$ of optimization. At the limit of $\beta \to \infty$, $p_\lambda^\beta$ converges to posterior $p_\lambda$. A lower bound of sampling efficiency $\text{eff}_\lambda^\beta \geq (Z_\lambda^\beta/\beta)^2$ could be established. More importantly, weight clipping always achieves smaller generalization bounds than optimization, and higher sampling efficiency than Bayesian learning.

**Theorem 3.** *For parameters $\beta \in (0, \infty)$, $D_{\text{KL}}(p_\lambda^\beta \| p_\lambda)$ and $\text{eff}_\lambda^\beta$ are both monotonically decreasing functions of $\beta$.*

## 7 Overparameterized Neural Network

The energy change $\Delta_f V(\theta)$ and entropy change $\Delta_f S(\theta)$ play a pivot role in previous sections. All quantities of interest, including KL divergence in the PAC-Bayesian bounds, weights in TransBL and sampling efficiency, depend on these two terms. Therefore, it is essential to understand the dynamics of these quantities. In this section, we will explore some simple infinitely wide neural network. In particular, this analysis will provide us insight for further developments in the deep Bayesian learning and PAC-Bayesian bounds.

### 7.1 Network Definition

We consider a feedforward network with $d$ fully connected layers. The parameter $\theta = \text{vec}(W_1, \ldots, W_d)$ is a vector of flattened weights. For a specific input $s_a$, the forward propagation of the network is defined as follows:

$$x_0(\theta, s_a) = s_a, \quad y_i(\theta, s_a) = c_i W_i x_{i-1}(\theta, s_a), \quad x_i(\theta, s_a) = \sigma_i(y_i(\theta, s_a)).$$

The output $y_d(\theta, s_a)$ is a scalar. The point-wise function $\sigma_i(y)$ is an activation function with bounded first to fourth order derivative. The hidden layer width at $i$-th layer is $n_i$ such that $y_i(\theta, s_a), x_i(\theta, s_a) \in \mathbb{R}^{n_i}$. $W_i \in \mathbb{R}^{n_i \times n_{i-1}}$ are trainable weights. The coefficient $c_i = 1/\sqrt{n_{i-1}}$ is used to ensure the output lies in a proper scale at infinite width limit, and is widely adopted in previous works [43, 51, 70, 40, 39, 3, 18, 59]. The network is initialized with Gaussian prior, such that $(W_i)_{j,k} \sim \mathcal{N}(0, 1)$.

We introduce the following notation to simplify the result. For a multi-variable scalar function $f(X)$, $\nabla_X f(X)$ is a row vector when $X$ is a vector. If $X$ is a matrix, then $(\nabla_X f(X))_{i,j} = \frac{\partial f(X)}{\partial X_{j,i}}$ and $\nabla_X f(X)$ share the same shape as $X^T$. For a multi-variable vector-value function $F(X)$, we define $(\nabla_X F(X))_{i,j} = \frac{\partial F_i(X)}{\partial X_j}$.

The network is trained by gradient flow or infinitesimal step size gradient descent. Let output vector be $(Y(\theta))_a = y_d(\theta, s_a)$ and total loss function as $\mathcal{L}(Y) = \sum_{a=1}^m l^s(Y_a, z_a)$, the gradient flow is defined as:

$$\frac{\text{d}}{\text{d}t}\theta(t) = -(\nabla_Y \mathcal{L}(Y(\theta(t)))\nabla_\theta Y(\theta(t)))^T.$$

### 7.2 Energy Term

The energy for the prior is defined as $V(\theta) = \sum_{i=1}^d \frac{1}{2}\|W_i\|_{\text{Fro}}^2$. We define $y_d^{(i)}(\theta, s_a) = \text{Tr}(\nabla_{W_i} y_d(\theta, s_a) W_i)$ and the dynamics of $V(\theta(t))$ is:

$$\frac{\text{d}}{\text{d}t}V(\theta(t)) = -\nabla_Y \mathcal{L}(Y(\theta(t))) \sum_{i=1}^d Y^{(i)}(\theta(t)). \tag{7}$$

We find that $y_d^{(i)}(\theta, s_a)$ can be computed by a group of auxiliary vector as Eq. (27) detailed in the appendix, and could be regarded as the tangent vector at output space by propagating a tangent vector $W_i$ from $i$-th layer.

For deriving the dynamics of $(Y^{(i)}(\theta))_a = y_d^{(i)}(\theta, s_a)$, we define $\Theta^{(i)}(\theta) = (\nabla_\theta Y^{(i)}(\theta))(\nabla_\theta Y(\theta))^T$ and it follows

$$\frac{\text{d}}{\text{d}t}Y^{(i)}(\theta(t)) = -\Theta^{(i)}(\theta(t))(\nabla_Y \mathcal{L}(Y(\theta(t))))^T. \tag{8}$$

We note that the definition of $Y^{(d)}$ follows $Y = Y^{(d)}$, therefore $\Theta^{(d)}$, as a gradient Gram matrix, is just neural tangent kernel (NTK) which has been studied in [39, 3, 18].

## 7.3 Entropy Term

We write down the gradient flow in an abstract form $\frac{d}{dt}\theta(t) = -g(\theta(t))$ and denote the solution as $\theta(t) = f(t, \theta(0)) = \theta(0) - \int_0^t g(\theta(t))dt$. This gradient flow is a special example of optimization flow in Section 3. Here the entropy change is $\Delta_t S(\theta) = \log J_{f(t,\cdot)}(\theta)$. According to Liouville formula, we have $\frac{d}{dt}(\Delta_t S(\theta(0))) = -\mathbf{div} g(\theta(t)) = -\operatorname{Tr}(\nabla g(\theta(t)))$. Given that the gradient has form $g(\theta) = -(\nabla_Y \mathcal{L}(Y(\theta))\nabla_\theta Y(\theta))^T$, the entropy change could be formulated as follows:

$$\frac{d}{dt}(\Delta_t S(\theta(0))) = -\operatorname{Tr}\left(\nabla_Y \nabla_Y \mathcal{L}(Y(\theta(t)))\Theta^{(d)}(\theta(t))\right) - \sum_{a=1}^m \nabla_{Y_a} \mathcal{L}(Y(\theta(t)))\operatorname{Tr}(H_a(\theta(t))).$$

(9)

The content in the first trace term in Eq. (9) is called Gauss-Newton matrix, which only represents curvature information of loss $l^s$. The matrix $H_a = \nabla_\theta \nabla_\theta Y_a$ is the Hessian of $y_d(\theta, s_a)$. It is known that spectral norm of Hessian vanishes for wide neural network [47]. However, we find that the trace is not constant and depends on initialization and training. For all $1 \le i \le j < k \le d, \alpha = 1, \ldots, n_j$, we introduce auxiliary vectors $(\xi^{(i,j)})_\alpha = c_i^2 \|x_{i-1}\|^2 \|\nabla_{y_i}(y_j)_\alpha\|^2$ and $\gamma_k^{(j)} = (\nabla_{x_j} y_k)\sigma_j''(y_j)$. Notice that we left out parameters $(\theta, s_a)$ for concision.

**Theorem 4.** *Let $\odot$ be element-wise product, we have*

$$\operatorname{Tr}(H_a(\theta(t))) = \sum_{j=1}^{d-1}(\nabla_{x_j} y_d)\left(\sigma_j''(y_j) \odot \left(\sum_{i=1}^j \xi^{(i,j)}\right)\right).$$

The above formula is a multi-variable version of $(f_{d-1} \circ \cdots \circ f_1)'' = \sum_{j=1}^{d-1}(f_{d-1} \circ \cdots \circ f_{j+1})' f_j''((f_{j-1} \circ \cdots \circ f_1)')^2$. $\xi^{(i,j)}$ plays the role of squared gradient and is a kind of diagonal NTK in middle layers.

We define $(\Gamma^{(i)}(\theta))_a = \gamma_d^{(i)}(\theta, s_a)$ and matrices $\Phi^{(i)}(\theta) = (\nabla_\theta \Gamma^{(i)}(\theta))(\nabla_\theta Y(\theta))^T$. Then we have for all $1 \le i < d$,

$$\frac{d}{dt}\Gamma^{(i)}(\theta(t)) = -\Phi^{(i)}(\theta(t))(\nabla_Y \mathcal{L}(Y(\theta(t))))^T.$$

(10)

## 7.4 Infinite Width Limit

The dynamics in previous two sections can be dramatically simplified in infinite width limit.

**Theorem 5.** *When hidden layers width $n = n_1, \ldots, n_{d-1}$ approaches infinity, the random vectors $Y^{(i)}(\theta(0))$ and $\Gamma^{(i)}(\theta(0))$ at initialization converges in law to a multivariate Gaussian distribution with zero mean and covariance matrix $\Sigma$. Moreover, the following quantities converge in probability to constant and don't change during finite training time:*

$$\Theta^{(i)}(\theta(t)) \to \Theta^{(i)}, \quad \Phi^{(i)}(\theta(t)) \to \Phi^{(i)}, \quad (\xi^{(i,j)}(\theta(t), s_a))_\alpha \to (\Xi^{(i,j)})_a.$$

(11)

*The specific formulas of $\Sigma$, $\Theta^{(i)}$, $\Phi^{(i)}$, and $\Xi^{(i,j)}$ are shown in Appendix D.*

The last line of above limits is for all $\alpha = 1, \ldots, n_j$, therefore theorem 4 could be simplified to:

$$\operatorname{Tr}(H_a(\theta(t))) = \sum_{j=1}^{d-1}(\Gamma^{(j)}(\theta(t)))_a \left(\sum_{i=1}^j (\Xi^{(i,j)})_a\right).$$

Theorem 5 demonstrates a function space picture, where the dynamics of $Y^{(i)}(\theta(t))$ and $\Gamma^{(i)}(\theta(t))$ only depend on these values themselves but not on internal dynamics of the network. The auxiliary vectors $Y^{(i)}(\theta(t))$ and $\Gamma^{(i)}(\theta(t))$ could be obtained by solving ODEs (8) and (10), $\frac{d}{dt}V$ and $\frac{d}{dt}S$ can be integrated according to Eqs. (7) and (9). Therefore, the KL divergence is accessible as an expectation in Eq. (1). Based on this KL divergence, we can obtain the final result of the PAC Bayes bound formulated as a function of training time and training data. The complete formula of this PAC Bayes bound is shown in Appendix F.4.

In order to get a sense of the how the generalization bound and sampling efficiency are influenced by training time, we show the asymptotic behaviour under mild assumption on loss function.

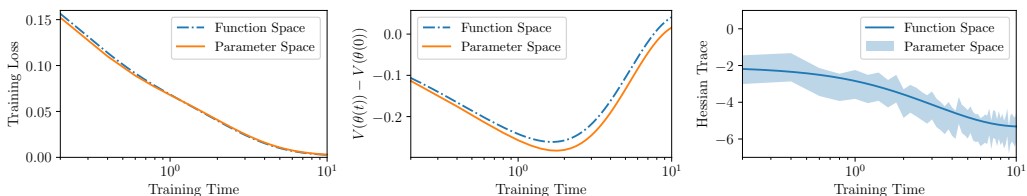

Figure 2: Comparison of dynamics in parameter space and function space.

**Corollary 6.** *If there exist $C_1$, $C_2$ such that $\left|\frac{\mathrm{d}}{\mathrm{d}y}l^s(y,t)\right| \leq C_1$, $\left|\frac{\mathrm{d}^2}{\mathrm{d}y^2}l^s(y,t)\right| \leq C_2$, then $D_{\mathrm{KL}}(q_t\|p) \leq \mathcal{O}(1)t + \mathcal{O}(1)t^2$, $\mathrm{eff}_\lambda \geq \mathcal{O}(1)\exp\left(-\mathcal{O}(1)t - \mathcal{O}(1)t^2\right)$ for any finite $t$, where $\mathcal{O}(1)$ represents positive constant irrelevant to $t$, $q_t$ is the output distribution of an infinitely wide network after training for $t$ length of time, and $\mathrm{eff}_\lambda$ is the sampling efficiency of importance sampling with $q_t$ as proposal distribution.*

# 8 Experiments

We first illustrate that Hessian trace doesn't vanish for overparameterized network and our analysis induces an efficient estimation of this value. Next, we verify our theoretical finding by comparing the dynamics of an overparameterized network in function space and parameter space. Finally, we demonstrate the interpolation of sampling and optimization.

## 8.1 Non-Diminishing Hessian Trace and Efficient Estimation

We consider a three hidden layers feedforward network with $2048$ as hidden layer width and $\tanh$ as nonlinear function.

The synthetic dataset is constructed on a unit circle. We choose $20$ values $\tau$ uniformly distributed between $0$ to $\pi$ and we let 2-dimensional input to the network be $s(\tau) = [\cos(\tau) \quad \sin(\tau)]$.

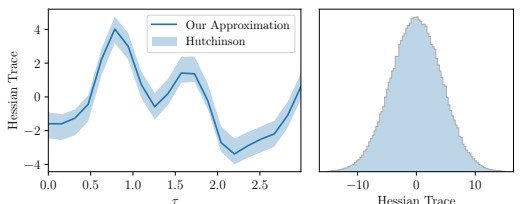

Figure 1: Hessian trace for one randomly initialized network (left) and the probability density of Hessian trace at initialization (right).

For each input $s(\tau)$, we are concerned about Hessian trace $\mathrm{Tr}(\nabla_\theta\nabla_\theta y_d(\theta, s(\tau)))$. Fast Hessian trace estimation by itself is a challenging problem, due to the high dimensional parameters and complex structure of the network. We follow previous works [36, 6, 78] and adopt Hutchinson's method to estimate the ground truth value of Hessian trace.

On the other hand, our analysis on the dynamics of overparameterized network reveals that certain factors which Hessian trace depends on are insensitive to initialization and training, therefore could be estimated by a fixed value to reduce computation. More details are shown in Appendix F.3. We apply the Hutchinson's method with 1000 independent random Rademacher vectors. The $3\sigma$ interval is plotted in the Figure 1. It is shown that our approximation aligns with ground truth well. Based on this approximation, the distribution of Hessian is also calculated in Figure 1.

## 8.2 Comparing the Dynamics in Parameter Space and Function Space

We use the same toy model as the previous section and set target values to be $t(\tau) = \frac{1}{2}\sin(3\tau)$ and loss to be mean squared error (MSE). For dynamics in parameter space, we run SGD with finite step size $0.01$ and mini-batch size $1$. For dynamics in function space, we solve Eqs. (7) to (10) with fixed matrices $\Theta^{(i)}$, $\Phi^{(i)}$ and $\Xi^{(i,j)}$. The result is plotted in Figure 2, where for SGD, the training time is defined as step size times iteration number. The ground truth of Hessian trace on one input is again estimated by Hutchinson's method. We can see from all three kinds of outputs that the function space

dynamics produces similar output as SGD. The error between them is attributed to discretization error for finite step size and finite width fluctuation.

### 8.3 Interpolation of Optimization and Bayesian Learning

We consider one-shot learning on Fashion-MNIST [75]. We randomly select two classes for binary classification and select one sample for each class as training dataset. We use a single hidden layer network with width being 1024 and softplus activation. We use loss $l(y, t) = 1/(1 + \exp(yt))$ and surrogate loss $l^s(y, t) = \log(1 + \exp(-yt))$ for gradient descent. For Gibbs measure, we fix $\lambda = 180$. The entropy change is approximately evaluated by integrating Eq. (9) with finite step size and fixed $\Theta^{(d)}$.

We train $10^5$ independent network and the results are shown in Figure 3. The shadow region represents $3\sigma$ interval. We first notice that during the training, the expectation of TransBL prediction is stable, yet the variance decreases, indicating the distribution of ensemble is becoming closer to Gibbs measure. This could be verified in middle figure in Figure 3, as the distribution of weights concentrates toward the partition function $\log Z_\lambda \approx -43.6$.

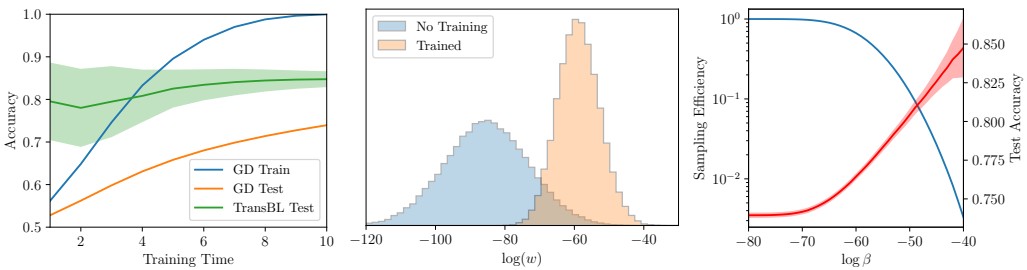

Figure 3: Test accuracy for ensemble trained by GD and TransBL with $\beta = \exp(-40)$ for weight clipping (left). Distribution of weights in TransBL (middle). Trade-off between sampling efficiency and test accuracy (right).

We also show a clear trade-off between sampling efficiency and test accuracy for TransBL with weight clipping, where the parameter $\beta$ plays a pivotal role in balancing these aspects. When $\beta$ is set high, the model leans towards a Bayesian posterior framework. Although this setting noticeably reduces the number of effective samples obtained, we observed an enhancement in the model's generalization ability, as depicted in the right figure of Figure 3. Conversely, with a smaller $\beta$, the model's behavior tends to resemble that of a typical deep ensemble, indicating a high sampling efficiency but worse generalization.

## 9 Conclusion

We study the generalization and sampling efficiency of Bayesian learning with special attention to overparameterized network. We show that both KL divergence, which governs generalization in PAC-Bayesian theory, and $\chi^2$ divergence, which determines sampling efficiency for importance sampling, are related to the change of energy and entropy loss during training. The dynamics of these two quantities in DNNs are studied, leading to a function space picture in infinite width limit. Our study also contributes to the understanding of DNNs Hessian trace, due to its involvement in entropy loss. By considering importance sampling with output distribution from gradient-based optimization as proposal distribution, we bridge the gap between optimization and Bayesian learning problems, and provide a flexible interpolation for accuracy-computation trade-off.

## Acknowledgement

This work was partially supported by NSF IIS 1838627, 1837956, 1956002, 2211492, CNS 2213701, CCF 2217003, DBI 2225775.

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

## A Related Works

**Generalization of Overparameterized Network**   Generalization for neural network has been studied based on stability [11, 12] norm [7, 57, 41, 27, 45], compression [4] , and special function class [5, 2]. [21, 81] achieves non-vacuous PAC-Bayesian generalization bounds based compression and perturbation robustness. Clerico et al. [15] derives PAC-Bayes bound and learning method for training a Gaussian process. A realizable PAC-Bayesian bound is also derived in [8] with sampling realizable networks instead of training by gradient descent. Mean field theory [22, 60, 66] could also be applied to obtain generalization bound.

**Early Stopping**   Restricting training time is a popular trick in deep learning. Related theoretical explanation includes a uniform stability approach [30], which related stability generalization bounds with training time, and a variational inference approach [19], which bound marginal likelihood with entropy. Compared to these works, our Corollary 6, although only applies for infinitely wide network, provides two new points of view. First we relates KL divergence with training time, thus provide a PAC-Bayes explanation for early stop. Second, we control the worst-case sampling efficiency with training time, thus provide a novel computational argument.

**Bayesian Deep Learning**   A comprehensive introduction of Bayesian Deep Learning and its relation to generalization is provided in [74]. It is known that sampling from the Gibbs measure is costly [37]. Bulk of work on Bayesian deep learning based on variational approximation [49, 33, 28, 9] has been developed. Despite the theoretical difficulty of exact Bayesian deep learning, various heuristic methods including SGD [48, 54], deep ensemble [23], and cyclic step size [80] have been shown to be successful in practice.

**Hessian of Overparameterized Network**   The convergence of various stochastic Hessian trace estimation methods are analyzed in [6]. There has been work focusing on empirical evaluation of Hessian of neural network [78]. Existing result in [47] shows that spectral norm of Hessian vanish in infinite width limit. Our paper contributes to this fields in two aspect. First, we show that Hessian trace doesn't vanish in infinite width limit. Second, we propose a novel Hessian trace estimation method in Appendix F.3, which is much more efficient than commonly used Hutchinson's method.

**MCMC**   : Markov Chain Monte Carlo (MCMC) utilize a kernel that leaves Bayesian posterior as stationary distribution. One source of bias of MCMC is finite number of iterations. The mixing time of MCMC suffer from curse of dimension, making it impractical to run MCMC till converge. Several orbit MCMC [68, 56] has been explored, seeking to incorporate non-equilibrium map into the transition kernel to reduce mixing time. They typically use damped Hamiltonian system or momentum gradient descent to bypass the entropy loss due to network curvature, thus suffer from increased $\chi^2$ divergence as discussed in Appendix F.2. Telescopic sum argument [26] with coupling technique [38, 32] have also been proposed to tackle this problem. Another source of bias is inaccurate implementation of transfer kernel. The discretization error of various Langevin dynamics method has been analyzed [1, 16, 64]. The bias from noisy loss function [62] and noisy gradient [72, 14] could be reduced but could not vanish. Papamarkou et al. [58] summarize more challenges of MCMC that leads to lack of convergence.

## B Discussion on the strength of assumption

We required $f$ to be a bijective and $J_f$ exist everywhere. This assumption is not as restrictive as it seems.

1. For gradient descent, $f(\theta) = \theta - h\nabla l$. If $l \in C^2$, $\nabla l$ is L-Lipschitz and $h \leq 1/L$, then $f$ is a diffeomorphism (Proposition 4.5 [44]), thus satisfying our assumption. In optimization literature, the above sufficient condition is mild and standard assumption to ensure the convergence of GD.

2. For gradient flow, $\frac{\mathrm{d}}{\mathrm{d}t}\theta = -\nabla l$. Our assumption immediately follows existence and uniqueness of ODE solution. For example, if $l \in C^2$ and $\nabla l$ is L-Lipschitz, then $f_t(\theta)$ is defined for all $t$ and $\theta$ (Corollary 2.6 [67]) and is a diffeomorphism (Theorem 2.10 [67]).

3. Momentum SGD and Adagrad always satisfy our assumption on enriched state space (Appendix F.2).

4. Same assumption was also used in previous research [50]. Moreover, many previous works use assumptions that imply our assumption as discussed in 1.

In Section 7, we only analyze the infinitesimal step-size. The main reason for this choice is that it allows the training dynamics of neural networks to be described by an ordinary differential equation (ODE), which simplifies the analysis. While it's possible that our approach could still be viable with discrete step sizes, it would make the theoretical analysis much more challenging. Practically, only discrete step sizes are implementable in experiments. However, our findings show that the theoretical predictions based on infinitesimal steps align closely with the experimental outcomes, as demonstrated in Figure 2.

One strong assumption imposed by this paper is an infinite width for the neural network. This assumption is vital for the feasibility of our theoretical analysis in Section 7 and is a common stance in numerous theoretical studies of neural networks [55, 73, 42, 51, 18, 39, 43, 76, 3]. The infinite-width perspective simplifies the dynamics and analysis but, admittedly, deviates from the practical, finite-width architectures typically used in real-world applications. Several researchers [29, 20, 35, 46, 63] have tried to understand the impact of finite width, known as "finite-width corrections". Delving into this adds complexity to the theoretical analysis, and hence it's left as a potential topic for future research. Moreover, we find that the dynamics in function space, derived from the theory of infinitely wide networks, closely resembles the behavior of actual finite-width networks, as can be seen in Figure 2.

## C    Additional Proofs

*Proof of Theorem 2.* First we show that sampling efficiency is determined by the $\chi^2$ divergence.

$$\frac{1}{\text{eff}_\lambda} = \frac{\mathbb{E}_{\theta \sim p}[w_\lambda^2(\theta)]}{Z_\lambda^2} = \frac{1}{Z_\lambda^2} \mathbb{E}_{\theta \sim q}\left[\left(e^{-\lambda r(f(\theta))} \frac{p(f(\theta))}{q(f(\theta))}\right)^2\right] = \mathbb{E}_{\theta \sim q}\left[\left(\frac{p_\lambda(\theta)}{q(\theta)}\right)^2\right] = D_{\chi^2}(p_\lambda \| q) + 1$$

(12)

The inequality in Eq. (6) is from Theorem 5 in [25]. □

*Proof of Theorem 3.* Based on some algebra and calculus, we have

$$(Z_\lambda^\beta)^2 \frac{\mathrm{d}}{\mathrm{d}\beta} D_{\text{KL}}(p_\lambda^\beta \| p_\lambda) = \left(\mathbb{E}_{\theta \sim p}[\log \frac{\beta}{w_\lambda(\theta)} \mathbb{1}(w_\lambda(\theta) \geq \beta)]\right) \times (\mathbb{E}_{\theta \sim p}[w_\lambda(\theta) \mathbb{1}(w_\lambda(\theta) < \beta)]) \leq 0$$

For sampling efficiency, we have

$$\mathbb{E}_{\theta \sim p}[v_\beta(w_\lambda(\theta))] \frac{\mathrm{d}}{\mathrm{d}\beta} \log \frac{1}{\text{eff}_\lambda^\beta} = 2\mathbb{E}_{\theta \sim p}[\mathbb{1}(w_\lambda(\theta) \geq \beta)] \times (\frac{\beta \mathbb{E}_{\theta \sim p}[v_\beta(w_\lambda(\theta))]}{\mathbb{E}_{\theta \sim p}[v_\beta(w_\lambda(\theta))^2]} - 1) \geq 0$$

□

*Proof of Eq. (27).* We discard the parameter $\theta, s_a$ without loss of generality.

$$\nabla_{W_i} y_d = c_i x_{i-1} \nabla_{y_i} y_d \tag{13}$$

$$\forall j \geq i, \ \nabla_{y_j} y_{j+1} y_j^{(i)} = c_{j+1} W_{j+1} \nabla_{y_j} x_j y_j^{(i)}$$
$$= c_{j+1} W_{j+1} \nabla_{y_j} \text{diag}(\sigma_j'(y_j)) y_j^{(i)} \tag{14}$$
$$= y_{j+1}^{(i)}$$

We finish the proof by combining above results.

$$
\begin{aligned}
\mathrm{Tr}(\nabla_{W_i} y_d W_i) &= \mathrm{Tr}(c_i x_{i-1} \nabla_{y_i} y_d W_i) \\
&= \mathrm{Tr}(\nabla_{y_i} y_d c_i W_i x_{i-1}) \\
&= \mathrm{Tr}\left(\nabla_{y_i} y_d y_i^{(i)}\right) \\
&= \mathrm{Tr}\left(\nabla_{y_{d-1}} y_d \dots \nabla_{y_i} y_{i+1} y_i^{(i)}\right) \\
&= y_d^{(i)}
\end{aligned}
\tag{15}
$$

$\square$

*Proof of Theorem 4.*

$$
\begin{aligned}
\nabla_{W_i} y_d &= c_i x_{i-1} \nabla_{y_i} y_d \\
&= (c_i x_{i-1})(\nabla_{y_{d-1}} y_d \dots \nabla_{y_i} y_{i+1}) \\
&= (c_i x_{i-1})((c_d W_d) \operatorname{diag}(\sigma'_{d-1}(y_{d-1})) \dots c_{i+1} W_{i+1} \operatorname{diag}(\sigma'_i(y_i)))
\end{aligned}
\tag{16}
$$

Only $\sigma'_j(y_j)$ for $i \le j < d$ depends on $W_i$ in the above formula.

$$
\nabla_{(W_i)_{\alpha,\beta}} y_d = (c_i x_{i-1})_\beta (\nabla_{x_j} y_d \operatorname{diag}(\sigma'_j(y_j)) \nabla_{(y_i)_\alpha} y_j)
\tag{17}
$$

$$
\begin{aligned}
\nabla_{(W_i)_{\alpha,\beta}} \nabla_{(W_i)_{\alpha,\beta}} y_d &= (c_i x_{i-1})_\beta \left( \sum_{j=i}^{d-1} (\nabla_{x_j} y_d)(\sigma''_j(y_j) \odot \nabla_{(W_i)_{\alpha,\beta}} y_j \odot \nabla_{(y_i)_\alpha} y_j) \right) \\
&= (c_i x_{i-1})_\beta^2 \left( \sum_{j=i}^{d-1} (\nabla_{x_j} y_d)(\sigma''_j(y_j) \odot \nabla_{(y_i)_\alpha} y_j \odot \nabla_{(y_i)_\alpha} y_j) \right)
\end{aligned}
\tag{18}
$$

$$
\sum_{\alpha=1}^{n_i} \sum_{\beta=1}^{n_{i-1}} \nabla_{(W_i)_{\alpha,\beta}} \nabla_{(W_i)_{\alpha,\beta}} y_d = \sum_{j=i}^{d-1} (\nabla_{x_j} y_d)(\sigma''_j(y_j) \odot \xi^{(i,j)})
\tag{19}
$$

$$
\begin{aligned}
\mathrm{Tr}\, H &= \sum_{i=1}^{d} \sum_{\alpha=1}^{n_i} \sum_{\beta=1}^{n_{i-1}} \nabla_{(W_i)_{\alpha,\beta}} \nabla_{(W_i)_{\alpha,\beta}} y_d \\
&= \sum_{i=1}^{d} \sum_{j=i}^{d-1} (\nabla_{x_j} y_d)(\sigma''_j(y_j) \odot \xi^{(i,j)}) \\
&= \sum_{j=1}^{d-1} (\nabla_{x_j} y_d)(\sigma''_j(y_j) \odot (\sum_{i=1}^{j} \xi^{(i,j)}))
\end{aligned}
\tag{20}
$$

$\square$

# D   Analysis of Infinitely Wide Network

We first give the analytic form of limits in Theorem 5.

The covariant matrix $\Sigma$ is defined recursively. We first define series of Gaussian random variables $y_k^{(i)}(s_a), \gamma_k^{(i)}(s_a)$, where variables with different subscript $k$ are independent. We also let $y_k(s_a) = y_k^{(d)}(s_a)$. The covariance of $y_k^{(i)}(s_a)$ and $y_k^{(j)}(s_b)$ are denoted as $\Sigma_k[y^{(i)}(s_a), y^{(j)}(s_b)]$. Similarly,

we can define $\Sigma_k[y^{(i)}(s_a), \gamma^{(j)}(s_b)]$ and $\Sigma_k[\gamma^{(i)}(s_a), \gamma^{(j)}(s_b)]$.

$$\Sigma_1[y^{(i)}(s_a), y^{(j)}(s_b)] = \Sigma_1[y^{(i)}(s_a), \gamma^{(j)}(s_b)] = \Sigma_1[\gamma^{(i)}(s_a), \gamma^{(j)}(s_b)] = s_a^T s_b/n_0$$

$$\Sigma_{k+1}[y^{(i)}(s_a), y^{(j)}(s_b)] = \mathbb{E}[I_{i,k}(y_k(s_a), y_k^{(i)}(s_a))I_{j,k}(y_k(s_b), y_k^{(j)}(s_b))]$$

$$\Sigma_{k+1}[y^{(i)}(s_a), \gamma^{(j)}(s_b)] = \mathbb{E}[I_{i,k}(y_k(s_a), y_k^{(i)}(s_a))J_{j,k}(y_k(s_b), \gamma_k^{(j)}(s_b))]$$

$$\Sigma_{k+1}[\gamma^{(i)}(s_a), \gamma^{(j)}(s_b)] = \mathbb{E}[J_{i,k}(y_k(s_a), \gamma_k^{(i)}(s_a))J_{j,k}(y_k(s_b), \gamma_k^{(j)}(s_b))]$$

$$I_{i,k}(y_k, y_k^{(i)}) = \begin{cases} \sigma_k(y_k) & i > k \\ \sigma_k'(y_k)y_k^{(i)} & i \le k \end{cases} \tag{21}$$

$$J_{i,k}(y_k, \gamma_k^{(i)}) = \begin{cases} \sigma_k(y_k) & i > k \\ \sigma_k''(y_k) & i = k \\ \sigma_k'(y_k)\gamma_k^{(i)} & i < k \end{cases}$$

In the infinite width limit, the covariance at initialization of $(Y^{(i)}(\theta(0)))_a$ and $(Y^{(j)}(\theta(0)))_b$ is $\Sigma_d[y^{(i)}(s_a), y^{(j)}(s_b)]$. Similarly, the covariance at initialization of $(Y^{(i)}(\theta(0)))_a$ and $(\Gamma^{(j)}(\theta(0)))_b$ is $\Sigma_d[y^{(i)}(s_a), \gamma^{(j)}(s_b)]$, the covariance at initialization of $(\Gamma^{(i)}(\theta(0)))_a$ and $(\Gamma^{(j)}(\theta(0)))_b$ is $\Sigma_d[\gamma^{(i)}(s_a), \gamma^{(j)}(s_b)]$.

Other matrices limits are

$$(\Theta^{(i)})_{a,b} = \sum_{j=1}^{d} \left( \prod_{k=j}^{d-1} \mathbb{E}[\sigma_k'(y_k(s_a))\sigma_k'(y_k(s_b))] \right) \Sigma_j[y^{(i)}(s_a), y(s_b)] \tag{22}$$

$$+ \sum_{j=1}^{d} \sum_{k=\max(i,j)}^{d-1} \left( \prod_{\substack{l=j \\ l \neq k}}^{d-1} \mathbb{E}[\sigma_l'(y_l(s_a))\sigma_l'(y_l(s_b))] \right) \mathbb{E}[y_k^{(i)}(s_a)\sigma_k''(y_k(s_a))\sigma_k'(y_k(s_b))]\Sigma_j[y(s_a), y(s_b)], \tag{23}$$

$$(\Phi^{(i)})_{a,b} = \sum_{j=i+1}^{d} \left( \prod_{k=j}^{d-1} \mathbb{E}[\sigma_k'(y_k(s_a))\sigma_k'(y_k(s_b))] \right) \Sigma_j[\gamma^{(i)}(s_a), y(s_b)]$$

$$+ \sum_{j=1}^{i} \left( \mathbb{E}[\sigma_i'''(y_i(s_a))\sigma_i'(y_i(s_b))] \prod_{\substack{k=j \\ k \neq i}}^{d-1} \mathbb{E}[\sigma_k'(y_k(s_a))\sigma_k'(y_k(s_b))] \right) \Sigma_j[y(s_a), y(s_b)] \tag{24}$$

$$+ \sum_{j=1}^{d} \sum_{k=\max(i+1,j)}^{d-1} \left( \prod_{\substack{l=j \\ l \neq k}}^{d-1} \mathbb{E}[\sigma_l'(y_l(s_a))\sigma_l'(y_l(s_b))] \right) \mathbb{E}[\gamma_k^{(i)}(s_a)\sigma_k''(y_k(s_a))\sigma_k'(y_k(s_b))]\Sigma_j[y(s_a), y(s_b)], \tag{25}$$

$$(\Xi^{(i,j)})_a = \left( \prod_{k=i}^{j-1} \mathbb{E}[\sigma_k'(y_k(s_a))\sigma_k'(y_k(s_a))] \right) \Sigma_i[y(s_a), y(s_a)]. \tag{26}$$

Before embarking on the proof, we first give an intuition of above results. We first notice that $y_d^{(i)}(\theta, s_a)$ can be defined recursively:

$$\forall j \le i \le d, \ y_j^{(i)}(\theta, s_a) = y_j(\theta, s_a),$$

$$\forall i \le j < d, \ y_{j+1}^{(i)}(\theta, s_a) = c_{j+1} W_{j+1} \operatorname{diag}(\sigma_j'(y_j(\theta, s_a)))y_j^{(i)}(\theta, s_a). \tag{27}$$

We can give a similar definition by forward propagation for $\gamma_j^{(i)}(\theta, s_a) = (\nabla_{x_i} y_j(\theta, s_a))\sigma_i''(y_i(\theta, s_a))$, which covers the case when $j \leq i$:

$$\forall j \leq i \leq d, \ \gamma_j^{(i)}(\theta, s_a) = y_j(\theta, s_a),$$
$$\gamma_{i+1}^{(i)}(\theta, s_a) = c_{i+1}W_{i+1}\sigma_i''(\gamma_i^{(i)}(\theta, s_a)), \tag{28}$$
$$\forall i < j < d, \ \gamma_{j+1}^{(i)}(\theta, s_a) = c_{j+1}W_{j+1}\,\mathrm{diag}(\sigma_j'(y_j(\theta, s_a)))\gamma_j^{(i)}(\theta, s_a).$$

Therefore, both $y_d^{(i)}$ and $\gamma_d^{(i)}$ can be regarded as output from some other "neural network". We compares these forward pass with original neural network as follows:

$$\forall i \leq j < d, \ y_{j+1}(\theta, s_a) = c_{j+1}W_{j+1}\sigma_j(y_j(\theta, s_a)). \tag{29}$$

Both $y_d^{(i)}$ and $\gamma_d^{(i)}$ propagate in a same way as original network until $i$-th layer. After that, $y_d^{(i)}$ propagate by linear transformation, with matrix $\mathrm{diag}(\sigma_j'(y_j(\theta, s_a)))$ being controlled by original network. $\gamma_d^{(i)}$ uses a different activation function $\sigma_i''$ which is different than original network with $\sigma_i$ in $i$-th layer, and propagate by linear transformation in a similar way as $y_d^{(i)}$.

When we consider the $y_d$, $y_d^{(i)}$ and $\gamma_d^{(i)}$ as output from a multiple branch neural network, the $\Sigma$ in Eq. (21) is just conjugate kernel, and $\Theta^{(i)}, \Phi^{(i)}, \Xi^{(i,j)}$ in Eq. (22) are fragment of NTK of the larger network.

*Proof of Theorem 5.* We first prove the convergence to Gaussian distribution. It follows the proof of convergence of $\Theta^{(i)}, \Phi^{(i)}, \Xi^{(i,j)}$. Finally we prove that these values don't change during training.

We note that in the forward propagation as shown in Eqs. (27) to (29), we only need to perform two kind of operations: matrix vector multiplication and map vector into matrix by $\mathrm{diag}$. The latter operation could be rewrite into element-wise product which is a non-linear element-wise vector operation, *e.g.* $\mathrm{diag}(\sigma_j'(y_j(\theta, s_a)))y_j^{(i)}(\theta, s_a) = \sigma_j'(y_j(\theta, s_a)) \odot y_j^{(i)}(\theta, s_a)$.

Therefore, the calculation of $y_d$, $y_d^{(i)}$ and $\gamma_d^{(i)}$ readily satisfies the standard of *Tensor Program* which is firstly introduced in [77] and further developed in [76]. Since we require the continuous polynomially bounded third order derivative for $\sigma$, the general result in [76] for *Tensor Program* could applies, and the convergence in law of outputs to Gaussian distribution and almost surely convergence of NTK at initialization is justified. What is left is the computation of the covariance and NTK at the infinite width limit.

The variance could be computed recursively. Given that the covariance matrix of each previous layer is established, and the convergence to Gaussian distribution of each element in previous layer, it is easy to verify the covariance matrix for next layer follows Eq. (21).

We next compute the NTK limit. Note that $\Theta^{(d)}$, as gradients inner product, is exactly the same NTK that has been studied in [39, 3, 18]. Moreover, $(\Xi^{(i,j)})_\alpha$ is just the diagonal NTK if we consider a sub neural network which coincide with original network in previous $j - 1$ layers, but use one single hidden unit $(y_j)_\alpha$ at $j$-th layer as output. Therefore, we only need to take care about $\Theta^{(i)}$ and $\Phi^{(i)}$ for $i < d$.

$$(\Theta^{(i)})_{a,b} = (\nabla_\theta y_d^{(i)}(s_a))^T \nabla_\theta y_d(s_b)$$
$$= \sum_{j=1}^d \mathrm{Tr}\left(\nabla_{W_j} y_d^{(i)}(s_a)^T \nabla_{W_j} y_d(s_b)\right) \tag{30}$$

$$\nabla_{W_j} y_d^{(i)}(s_a) = c_j x_{j-1}^{(i)}(s_a)\nabla_{y_j} y_d^{(i)}(s_a) + \sum_{k=\max(i,j)}^{d-1} c_j x_{j-1}(s_a)(\nabla_{x_k} y_d(s_a)^T \odot y_k^{(i)}(s_a) \odot \sigma_k''(y_k(s_a)))^T \nabla_{y_j} y_k(s_a)$$
$$\tag{31}$$

The first term is easy to calculate and shares similar form with classical NTK after computing the trace $\mathrm{Tr}\left(\nabla_{W_j} y_d^{(i)}(s_a)^T \nabla_{W_j} y_d(s_b)\right)$.

$$\nabla_{y_j} y_d^{(i)}(s_a) = c_d W_d\,\mathrm{diag}(\sigma_{i-1}'(y_{i-1}(s_a)))\dots c_{j+1}W_{j+1}\,\mathrm{diag}(\sigma_j'(y_j(s_a))) = \nabla_{y_j} y_d(s_a) \tag{32}$$

Combining above results, we have

$$(\Theta^{(i)})_{a,b} = \sum_{j=1}^{d} \nabla_{y_j} y_d(s_a) \nabla_{y_j} y_d(s_b)^T c_j^2(x_{j-1}^{(i)}(s_a) x_{j-1}(s_b)) \tag{33}$$

The limit of first term is $\overset{\text{a.s.}}{\lim} \nabla_{y_j} y_d(s_a) \nabla_{y_j} y_d(s_b)^T = \prod_{k=j}^{d-1} \mathbb{E}[\sigma_k'(y_k(s_a))\sigma_k'(y_k(s_b))]$. In order to calculate the second term $c_j^2(x_{j-1}^{(i)}(s_a) x_{j-1}(s_b))$, we note that $\mathbb{E}_{w \sim \mathcal{N}(0,I)}[ww^T] = I$ and $(W_j)_\alpha$ is a standard normal random vector, then we have

$$c_j^2(x_{j-1}^{(i)}(s_a) x_{j-1}(s_b)) = \mathbb{E}[(c_j(W_j)_\alpha x_{j-1}^{(i)}(s_a))^T c_j(W_j)_\alpha x_{j-1}(s_b)] = \mathbb{E}[c_j^2(y_j^{(i)}(s_a) y_j(s_b))]$$

so the limit is $\Sigma_j[y^{(i)}(s_a), y(s_b)]$.

All other terms comes from second order derivative of non-linear unit $\sigma_k$ and doesn't show up in classical NTK result. We note in $\text{Tr}\Big((c_j x_{j-1}(s_a)(\nabla_{x_k} y_d(s_a)^T \odot y_k^{(i)}(s_a) \odot \sigma_k''(y_k(s_a)))^T \nabla_{y_j} y_k(s_a))^T \nabla_{W_j} y_d(s_b)\Big)$ = $(c_j^2 x_{j-1}(s_a)^T x_{j-1}(s_b))(\nabla_{y_j} y_d(s_b) \nabla_{y_j} y_k(s_a)^T (\nabla_{x_k} y_d(s_a)^T \odot y_k^{(i)}(s_a) \odot \sigma_k''(y_k(s_a))))$, $c_j^2 x_{j-1}(s_a)^T x_{j-1}(s_b) = \mathcal{O}(1)$ converge to a fixed value, in $\nabla_{y_j} y_d(s_b) \nabla_{y_j} y_k(s_a)^T \odot \nabla_{x_k} y_d(s_a) = \nabla_{x_k} y_d(s_b) \text{diag } \sigma_k'(y_k(s_a)) \nabla_{y_j} y_k(s_b) \nabla_{y_j} y_k(s_a)^T \odot \nabla_{x_k} y_d(s_a)$, $\nabla_{y_j} y_k(s_b) \nabla_{y_j} y_k(s_a)^T$ converges to fixed values times identity matrix, and $\nabla_{x_k} y_d(s_b)^T \nabla_{x_k} y_d(s_b)$ also converges to fixed values times identity matrix. The rest terms converge to its expectation which contains $y_k^{(i)}(s_a)$, $\sigma_k''(y_k(s_a))$ and $\sigma_k'(y_k(s_b))$.

$$\begin{aligned}(\Phi^{(i)})_{a,b} &= (\nabla_\theta \gamma_d^{(i)}(s_a))^T \nabla_\theta y_d(s_b) \\ &= \sum_{j=1}^{i} \text{Tr}\Big(\nabla_{W_j} \gamma_d^{(i)}(s_a)^T \nabla_{W_j} y_d(s_b)\Big) + \sum_{j=i+1}^{d} \text{Tr}\Big(\nabla_{W_j} \gamma_d^{(i)}(s_a)^T \nabla_{W_j} y_d(s_b)\Big)\end{aligned} \tag{34}$$

Note that $\gamma_d^{(i)}(\theta, s_a) = (\nabla_{x_i} y_d(\theta, s_a))\sigma_i''(y_i(\theta, s_a))$ and when $j \leq i$ the value $y_i(\theta, s_a)$ depends on $W_i$. Therefore we conduct a separate discussion. When $j > i$, the limit value could be derived in a way similar to $(\Theta^{(i)})_{a,b}$. For $j \leq i$,

$$\begin{aligned}\nabla_{W_j} \gamma_d^{(i)}(s_a) &= c_j x_{j-1}^{(i)}(s_a) \nabla_{y_j} \gamma_d^{(i)}(s_a) \\ &\quad + \sum_{k=j}^{d-1} c_j x_{j-1}(s_a)(\nabla_{x_k} y_d(s_a)^T \odot \gamma_k^{(i)}(s_a) \odot \sigma_k''(y_k(s_a)))^T \nabla_{y_j} y_k(s_a)\end{aligned} \tag{35}$$

The summations are again from the second order derivative of non-linear unit and the limit could be derived in a similar way to $\Phi^{(i)}$. We focus on first term is somewhat different since it depends on third order derivative of non-linear unit.

$$\nabla_{y_j} \gamma_d^{(i)}(s_a) = c_d W_d \text{diag}(\sigma_{i-1}'(y_{i-1}(s_a))) \ldots c_{i+1} W_{i+1} \text{diag}(\sigma_i'''(y_i(s_a))) \ldots \ldots c_{j+1} W_{j+1} \text{diag}(\sigma_j'(y_j(s_a))) \tag{36}$$

$\nabla_{y_j} \gamma_d^{(i)}(s_a)$ differs from $\nabla_{y_j} y_d^{(i)}(s_a)$ and $\nabla_{y_j} y_d(s_a)$ in one term $\sigma_i'''(y_i(s_a))$ and that fact is reflected in Eq. (25).

Finally we prove that $\Theta^{(i)}, \Phi^{(i)}, \Xi^{(i,j)}$ are constant during training. Since these values are all inner product of gradients, we just need to show gradients $\nabla_{W_j} y_d(\theta, s_a)$, $\nabla_{W_j} y_d^{(i)}(\theta, s_a)$ and $\nabla_{W_j} \gamma_d^{(i)}(\theta, s_a)$ doesn't change too much. We write down the formula of these gradient without the

summation terms which could be controlled similarly.

$$\nabla_{W_j} y_d(\theta, s_a) = c_j \sigma_{j-1}(y_{j-1}(\theta, s_a)) \nabla_{y_j} y_d(\theta, s_a)$$

$$\nabla_{W_j} y_d^{(i)}(\theta, s_a) = c_j \sigma'_{j-1}(y_{j-1}(\theta, s_a)) \odot y_{j-1}^{(i)}(\theta, s_a) \nabla_{y_j} y_d(\theta, s_a) \qquad \forall j > i$$

$$\nabla_{W_j} y_d^{(i)}(\theta, s_a) = c_j \sigma_{j-1}(y_{j-1}(\theta, s_a)) \nabla_{y_j} y_d(\theta, s_a) \qquad \forall j \le i$$

$$\nabla_{W_j} \gamma_d^{(i)}(\theta, s_a) = c_j \sigma'_{j-1}(y_{j-1}(\theta, s_a)) \odot \gamma_{j-1}^{(i)}(\theta, s_a) \nabla_{y_j} y_d(\theta, s_a) \qquad \forall j > i+1 \qquad (37)$$

$$\nabla_{W_j} \gamma_d^{(i)}(\theta, s_a) = c_j \sigma''_{j-1}(y_{j-1}(\theta, s_a)) \nabla_{y_j} y_d(\theta, s_a) \qquad \forall j = i+1$$

$$\nabla_{W_j} \gamma_d^{(i)}(\theta, s_a) = c_j \sigma_{j-1}(y_{j-1}(\theta, s_a)) \nabla_{y_{i+1}} y_d(\theta, s_a)(c_{i+1} W_{i+1}) \times \qquad \forall j < i+1$$
$$\operatorname{diag}(\sigma'''_i(y_i(\theta, s_a))) \nabla_{y_j} y_i(\theta, s_a)$$

We rely on following Local Lipschitz property to prove the stability of gradients during training.

**Lemma 7** (Local Lipschitz Condition). *Let $v_{j-1}(\theta)$ to denote any of following values:*

$$\sigma_{j-1}(y_{j-1}(\theta, s_a)),$$
$$\sigma'_{j-1}(y_{j-1}(\theta, s_a)) \odot y_{j-1}^{(i)}(\theta, s_a),$$
$$\sigma'_{j-1}(y_{j-1}(\theta, s_a)) \odot \gamma_{j-1}^{(i)}(\theta, s_a), \qquad (38)$$
$$\sigma''_{j-1}(y_{j-1}(\theta, s_a)),$$

*there is a constant $K > 0$ such that for every $R > 0$, with probability increasing to 1 at limit of $n \to \infty$, or equivalently, with high probability over random initialization (w.h.p.o.r.i.) of $\theta$, for any $\theta_1, \theta_2 \in B(\theta, R)$, where $B(\theta, R) = \{\theta' \mid \|\theta' - \theta\| < R\}$, the following holds*

$$\|v_j(\theta_1) - v_j(\theta_2)\| \le K \|\theta_1 - \theta_2\|,$$
$$\|v_j(\theta_1)\| \le K \sqrt{n}. \qquad (39)$$

*Moreover, let $v_j(\theta)$ to denote any of following values:*

$$\nabla_{y_j} y_d(\theta, s_a),$$
$$\nabla_{y_{i+1}} y_d(\theta, s_a)(c_{i+1} W_{i+1}) \operatorname{diag}(\sigma'''_i(y_i(\theta, s_a))) \nabla_{y_j} y_i(\theta, s_a), \qquad (40)$$

*there is a constant $K > 0$ such that for every $R > 0$, w.h.p.o.r.i. of $\theta$, for any $\theta_1, \theta_2 \in B(\theta, R)$,*

$$\|v_j(\theta_1) - v_j(\theta_2)\| \le \frac{1}{\sqrt{n}} K \|\theta_1 - \theta_2\|,$$
$$\|v_j(\theta_1)\| \le K. \qquad (41)$$

By using the norm upper bound in Lemma 7, we have that for any $R > 0$ and $\theta \in B(\theta(0), R)$, the gradient norm $\operatorname{Tr}\left((\nabla_{W_j} y_d(\theta, s_a))^T \nabla_{W_j} y_d(\theta, s_a)\right) = c_j^2 \|\sigma_{j-1}(y_{j-1}(\theta, s_a))\|^2 \|\nabla_{y_j} y_d(\theta, s_a)\|^2 \le K$ is bounded w.h.p.o.r.i. . For any finite training time, $R$ could be selected large enough to ensure $\theta(t) \in B(\theta(0), R)$ during training. The norm bound of difference in Lemma 7 also ensures that the change of gradient norm is bounded by $\frac{1}{\sqrt{n}} K \|\theta(t) - \theta(0)\|$ in $B(\theta(0), R)$ w.h.p.o.r.i. which is diminishing at the limit $n \to 0$.

$\square$

*Proof of Lemma 7.* All vectors $v_j$ in Lemma 7 could be computed sequentially by three kinds of operations: matrix multiplication $\zeta = W\varepsilon$, element-wise non-linearity $\varepsilon = \phi(\zeta)$ with $\phi$ having bounded derivative and element-wise multiplication $\varepsilon = \phi(\zeta) \odot \zeta'$, where the element-wise non-linear function $\phi$ is bounded and Lipschitz.

For the first kind of operation, given that $W$ follows standard normal distribution, we have the $\zeta$ is also normally distributed and enjoy following high probability bound.

**Eq. (2.19) of [71]**  For $n$ dimensional standard Gaussian random vector $\zeta \sim \mathcal{N}(0, I_n)$, for any $t$

$$\Pr\Big(\|\zeta\|^2 \geq n + t\Big) \leq 2e^{-nt^2/8}. \tag{42}$$

Therefore we have that for $\zeta = W\varepsilon$, $\|\zeta\| \leq K\sqrt{n}\|\varepsilon\|$ for some $K$ with high probability.

For element-wise non-linearity $\varepsilon = \phi(\zeta)$ where $\zeta$ is a standard Gaussian random vector with i.i.d. coordinates, we derive the Chernoff bound as follows.

$$
\begin{aligned}
\Pr\big(\|\phi(\zeta)\| \geq K\sqrt{n}\big) &= \Pr\left(\sum_{i=1}^{n} \phi(\zeta_i)^2 \geq K^2 n\right) \\
&= \inf_{t \geq 0} \Pr\left(\exp\left(t\sum_{i=1}^{n}\phi(\zeta_i)^2\right) \geq \exp\big(tK^2 n\big)\right) \\
&\leq \inf_{t \geq 0} \frac{\mathbb{E}[\exp\big(t\sum_{i=1}^{n}\phi(\zeta_i)^2)\big)]}{e^{tK^2 n}} \\
&= \inf_{t \geq 0} e^{-tK^2 n} \prod_{i=1}^{n} \mathbb{E}[\exp\big(t\phi(\zeta_i)^2\big))] \\
&\leq \inf_{t \geq 0} e^{-tK^2 n} \prod_{i=1}^{n} \mathbb{E}[\exp\big(t(\mathcal{O}(1) + \mathcal{O}(1)\zeta_i^2)\big))] \\
&\leq e^{-\mathcal{O}(1)K^2 n + \mathcal{O}(1)n}
\end{aligned}
\tag{43}
$$

Therefore, we have $\|\varepsilon\| = \|\phi(\zeta)\| \leq K\sqrt{n}$ w.h.p.o.r.i. for large enough $K$.

For element-wise multiplication $\varepsilon = \phi(\zeta) \odot \zeta'$, since $\phi$ is bounded, $\|\varepsilon\| \leq \mathcal{O}(1)\|\zeta'\|$ all the time.

Recursively apply above three operations gives the bound of norm $\|v_j\|$.

In order to control the norm of difference, we note that for matrix multiplication:

$$
\begin{aligned}
\|W_1\varepsilon_1 - W_2\varepsilon_2\| &= \|W_1\varepsilon_1 - W_1\varepsilon_2 + W_1\varepsilon_2 - W_2\varepsilon_2\| \\
&\leq \|W_1(\varepsilon_1 - \varepsilon_2)\| + \|W_1 - W_2\|_{\text{op}}\|\varepsilon_2\| \\
&\leq \|W_1(\varepsilon_1 - \varepsilon_2)\| + \|W_1 - W_2\|_{\text{Fro}}\|\varepsilon_2\|
\end{aligned}
\tag{44}
$$

The first term could be controlled with high probability as $\|W_1(\varepsilon_1 - \varepsilon_2)\| \leq \|W_0(\varepsilon_1 - \varepsilon_2)\| + \|(W_1 - W_0)(\varepsilon_1 - \varepsilon_2)\| \leq K\sqrt{n}\|\varepsilon_1 - \varepsilon_2\| + R\|\varepsilon_1 - \varepsilon_2\| \leq \mathcal{O}(1)\sqrt{n}\|\varepsilon_1 - \varepsilon_2\|$ for large enough $n$, and $W_0$ is the parameter at initialization, $R$ is the maximum distance between $W_1$ and $W_0$.

For element-wise non-linearity $\varepsilon = \phi(\zeta)$, since $\phi$ is Lipschitz, we have the difference as

$$\|\varepsilon_1 - \varepsilon_2\| = \|\phi(\zeta_1) - \phi(\zeta_2)\| \leq \mathcal{O}(1)\|\zeta_1 - \zeta_2\| \tag{45}$$

For element-wise multiplication $\varepsilon = \phi(\zeta) \odot \zeta'$, since $\phi$ is bounded and Lipschitz, we have the difference as

$$
\begin{aligned}
\|\varepsilon_1 - \varepsilon_2\| &= \|\phi(\zeta_1) \odot \zeta'_1 - \phi(\zeta_2) \odot \zeta'_2\| \\
&\leq \|(\phi(\zeta_1) - \phi(\zeta_2)) \odot \zeta'_1\| + \|\phi(\zeta_2) \odot (\zeta'_1 - \zeta'_2)\| \\
&\leq \mathcal{O}(1)\|(\zeta_1 - \zeta_2) \odot \zeta'_1\| + \mathcal{O}(1)\|\zeta'_1 - \zeta'_2\| \\
&\leq \mathcal{O}(1)\|\zeta_1 - \zeta_2\| + \mathcal{O}(1)\|\zeta'_1 - \zeta'_2\|
\end{aligned}
\tag{46}
$$

The first term is reduced by noticing $\zeta'_1$ has $\mathcal{O}(1)$ elements.

Recursively apply above three operations gives the bound of norm of difference.

$\square$

*Proof of Corollary 6.*  According to Theorem 5, in the infinite width limit, we have

$$
\begin{aligned}
\frac{\mathrm{d}}{\mathrm{d}t}\left(\sum_{i=1}^{d} Y^{(i)}(\theta(t))\right) &= -(\sum_{i=1}^{d}\Theta^{(i)})(\nabla_Y\mathcal{L})^T \\
\frac{\mathrm{d}}{\mathrm{d}t}\left(\sum_{i=1}^{d-1}\Gamma^{(i)}(\theta(t))\right) &= -(\sum_{i=1}^{d-1}\Phi^{(i)})(\nabla_Y\mathcal{L})^T
\end{aligned}
\tag{47}
$$

We use $\mathcal{O}(1)$ to denote constants that is irrelevant to t. Given that $\nabla_Y \mathcal{L}$ are bounded, we have

$$
\begin{aligned}
\left\| \sum_{i=1}^{d} Y^{(i)}(\theta(t)) \right\| &\leq \left\| \sum_{i=1}^{d} Y^{(i)}(\theta(0)) \right\| + \mathcal{O}(1)t \\
\left\| \sum_{i=1}^{d-1} \Gamma^{(i)}(\theta(t)) \right\| &\leq \left\| \sum_{i=1}^{d-1} \Gamma^{(i)}(\theta(0)) \right\| + \mathcal{O}(1)t
\end{aligned}
\tag{48}
$$

It follows that the Hessian is also bounded.

$$
\mathrm{Tr}\{H_a(\theta(t))\} \leq \mathcal{O}(1)\left\| \sum_{i=1}^{d-1} \Gamma^{(i)}(\theta(0)) \right\| + \mathcal{O}(1)t
\tag{49}
$$

The time derivative and finite time change of energy and entropy is

$$
\begin{aligned}
\left| \frac{\mathrm{d}}{\mathrm{d}t} V(\theta(t)) \right| &\leq \mathcal{O}(1)\left\| \sum_{i=1}^{d} Y^{(i)}(\theta(0)) \right\| + \mathcal{O}(1)t \\
|\Delta_t V(\theta(0))| = |V(\theta(t)) - V(\theta(0))| &\leq \mathcal{O}(1)\left\| \sum_{i=1}^{d} Y^{(i)}(\theta(0)) \right\| t + \mathcal{O}(1)t^2 \\
\left| \frac{\mathrm{d}}{\mathrm{d}t}(\Delta_t S(\theta(0))) \right| &\leq \mathcal{O}(1) + \mathcal{O}(1)\left\| \sum_{i=1}^{d-1} \Gamma^{(i)}(\theta(0)) \right\| + \mathcal{O}(1)t \\
|\Delta_t S(\theta(0))| &\leq \mathcal{O}(1)\left\| \sum_{i=1}^{d-1} \Gamma^{(i)}(\theta(0)) \right\| t + \mathcal{O}(1)t + \mathcal{O}(1)t^2
\end{aligned}
\tag{50}
$$

The KL divergence is

$$
\begin{aligned}
D_{\mathrm{KL}}(q_t \| p) =& \mathbb{E}[\Delta_t V(\theta(0)) - \Delta_t S(\theta(0))] \\
\leq& \mathbb{E}\left[ \mathcal{O}(1)\left\| \sum_{i=1}^{d} Y^{(i)}(\theta(0)) \right\| t + \mathcal{O}(1)\left\| \sum_{i=1}^{d-1} \Gamma^{(i)}(\theta(0)) \right\| t + \mathcal{O}(1)t + \mathcal{O}(1)t^2 \right] \\
\leq& \mathcal{O}(1)t + \mathcal{O}(1)t^2
\end{aligned}
\tag{51}
$$

The last inequality is because $\sum_{i=1}^{d} Y^{(i)}(\theta(0))$ follows Gaussian distribution and the expectation of norm is finite.

The sampling efficiency could also be controlled.

$$
\begin{aligned}
\frac{1}{\mathrm{eff}_\lambda} =& \frac{1}{Z_\lambda^2} \mathbb{E}[\exp(-2\lambda r(\theta(t)) - 2\Delta_t V(\theta(0)) + 2\Delta_t S(\theta(0)))] \\
\leq& \frac{1}{Z_\lambda^2} \mathbb{E}[\exp(-2\Delta_t V(\theta(0)) + 2\Delta_t S(\theta(0)))] \\
\leq& \mathcal{O}(1)\mathbb{E}\left[ \exp\left( \mathcal{O}(1)\left\| \sum_{i=1}^{d} Y^{(i)}(\theta(0)) \right\| t + \mathcal{O}(1)\left\| \sum_{i=1}^{d-1} \Gamma^{(i)}(\theta(0)) \right\| t + \mathcal{O}(1)t + \mathcal{O}(1)t^2 \right) \right] \\
\leq& \mathcal{O}(1)\exp\left( \mathcal{O}(1)t + \mathcal{O}(1)t^2 \right) \mathbb{E}\left[ \exp\left( \mathcal{O}(1)\left\| \begin{bmatrix} \sum_{i=1}^{d} Y^{(i)}(\theta(0)) \\ \sum_{i=1}^{d-1} \Gamma^{(i)}(\theta(0)) \end{bmatrix} \right\| t \right) \right] \\
\leq& \mathcal{O}(1)\exp\left( \mathcal{O}(1)t + \mathcal{O}(1)t^2 \right)
\end{aligned}
\tag{52}
$$

The first inequality comes from $r(\theta) \geq 0$. The last inequality follows from the fact that, for random vectors with tail similar to log-normal distribution, the expectation exists. More specifically, let $\sigma_i(\Sigma)$ is eigenvalues of covariance matrix $\Sigma$, we have $\mathbb{E}_{x \sim \mathcal{N}(0,\Sigma)}[\exp(c\|x\|)] \leq \prod_i \mathbb{E}_{y_i \sim \mathcal{N}(0,\sigma_i(\Sigma))}[\exp(c\|y_i\|)] = \prod_i \exp\left(c^2 \sigma_i^2(\Sigma)/2\right)(1 + \mathrm{erf}(c\sigma_i(\Sigma)/2))$. $\qquad\square$

## E  Additional Experiment Results

In Figure 4, we show the correlation between KL divergence and generalization gap during training. We also show how weight distribution evolves.

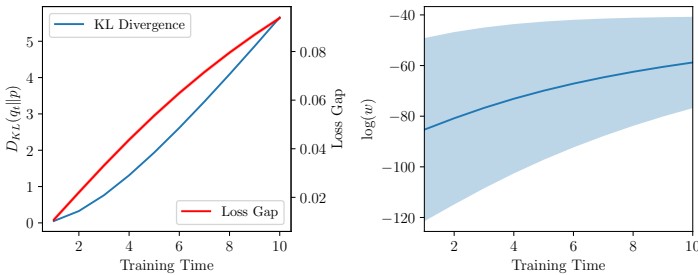

Figure 4: The KL divergence, training and testing loss gap, and time dependence of weight distribution of experiment in Section 8.3.

# F    Additional Discussion on Algorithms

## F.1    An illustrative example of TransBL

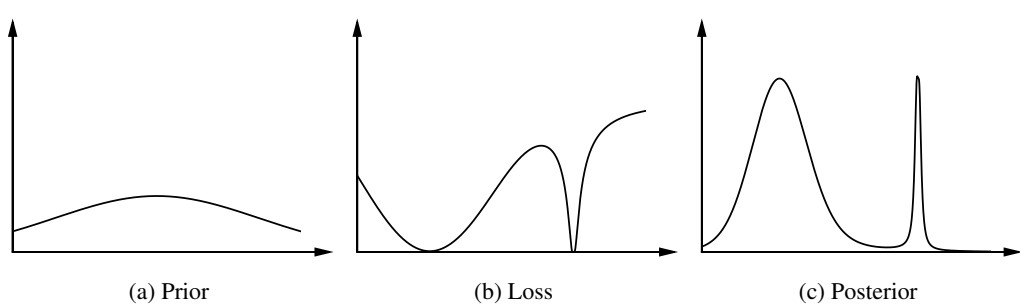

(a) Prior            (b) Loss            (c) Posterior

Figure 5: Bayesian learning process

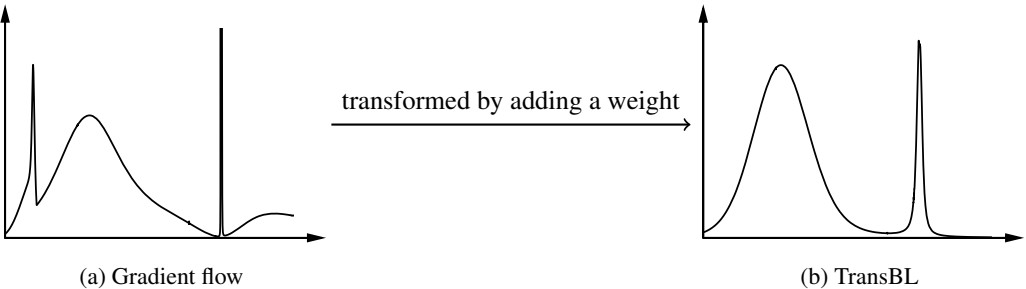

(a) Gradient flow            (b) TransBL

Figure 6: Illsutration of TransBL

In the presented motivating example, we explore a univariate loss function that presents two distinct global minima with zero loss. One of these is characterized as a sharp minimum, while the other represents a flat minimum. If the function were to be randomly initialized and then optimized, it might converge to either of the two localities. However, insights from PAC Bayesian indicate that the flat minimum is surrounded by a higher posterior probability density. A direct initialization from the prior, followed by training using gradient flow, often results in a significant deviation of the ensemble from the posterior. This is primarily because the optimization process fails to recognize the presence of a sharp minimum. The intuitive approach of the TransBL method is to apply a small weight to the solution found within the sharp minimum. Consequently, TransBL can adeptly recreate the posterior, as depicted in the Figure 6b.

Furthermore, we demonstrate the interpolation between the ensemble distribution obtained from optimization and Bayesian posterior through weight clipping in Section 6.1. In Figure 7a, with $\beta = 0.2$, the curve leans more towards the optimization result, revealing a broader spread and

less-defined peaks. This shows a scenario where optimization has a stronger influence than Bayesian learning. Yet with $\beta = 5$, the distribution is almost akin to what one would expect from a Bayesian posterior. In essence, the parameter $\beta$ serves as a tuning knob, allowing us to interpolate and traverse the spectrum from pure optimization-driven results to outcomes heavily influenced by Bayesian learning. This interpolation mechanism offers a flexible approach to merge the strengths of both methodologies.

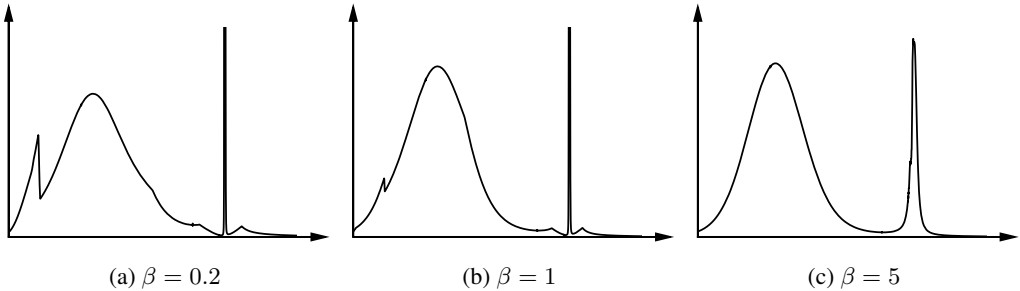

(a) $\beta = 0.2$        (b) $\beta = 1$        (c) $\beta = 5$

Figure 7: Interpolation between optimization and Bayesian learning

### F.2 Optimization Algorithms

In the main paper, we analyze the change of energy $\Delta V$ and entropy $\Delta S$ for gradient descent (GD). However, with finite step size, GD might not be reversible. Another difficulty lies in the computation of entropy change. As shown in Eq. (9), the entropy change is related to the curvature of loss $l^s$ and Hessian trace of neural network, where the second part is hard to compute exactly.

We note that this problem could be alleviated for algorithms with enriched state space, but at a price of lower sampling efficiency.

**Momentum SGD** For parameter $x$ and momentum $v$, the Momentum SGD with fraction $\gamma$, step size $h$ and gradient oracle $g$ can be formulated into following two steps:

$$\begin{cases} v \leftarrow \gamma v + g(x) \\ x \leftarrow x - v \end{cases} . \tag{53}$$

By defining $y = [x \quad v]$, Momentum SGD turns into a transform $T(y)$ composed by two sub transforms $T(y) = L_2 \circ L_1(y)$ that correspond to two lines separately.

The determinant of Jacobian for $L_1$ is $\det(\gamma I) = \gamma^d$, where $d$ is dimension of parameters. The determinant of Jacobian for $L_2$ is 1, therefore, the determinant of Jacobian for Momentum SGD is $\gamma^d$.

**Adagrad** For parameter $x$, first-order momentum $v$ and second-order momentum $m$, the Adagrad is:

$$\begin{cases} m \leftarrow m + g(x)^2 \\ v \leftarrow \gamma v + (1 - \gamma)\frac{g(x)}{\sqrt{m}} \\ x \leftarrow x - hv \end{cases} . \tag{54}$$

The square, square root and division in above formula are all element-wise.

Similar to Momentum SGD, the whole transform of Adagrad could be decomposed into three sub transform. The first and third step are both volume-preserving, and second step compress momentum by $\gamma$ in each dimension. Therefore the determinant of Jacobian for Adagrad is $\gamma^d$.

Both above two algorithms enjoy simple form of Jacobian determinant. However, the sampling efficiency degenerates since $\chi^2$ divergence increases when we consider joint probability distribution of parameter and auxiliary variables, like momentum.

### F.3 Approximate DNN Hessian Trace by Forward Propagation

We first recall that due to Theorem 4, we have

$$\text{Tr}(H_a(\theta(t))) = \sum_{j=1}^{d-1} (\nabla_{x_j} y_d) \left( \sigma_j''(y_j) \odot \left( \sum_{i=1}^{j} \xi^{(i,j)} \right) \right).$$

Our motivation is based on the stability of $\xi^{(i,j)}$. We have shown in Theorem 5 that in the infinite width limit, $\xi^{(i,j)}$ converge to a vector with same elements and is stable during training. For the wide but finite width network, this property is largely preserved.

If we replace $\xi^{(i,j)}$ with its limit value, we obtain following formula:

$$\text{Tr}(H_a(\theta(t))) \approx \sum_{j=1}^{d-1} (\nabla_{x_j} y_d(\theta(t), s_a)) \sigma_j''(y_j(\theta(t), s_a)) \left( \sum_{i=1}^{j} (\Xi^{(i,j)})_a \right).$$

In order to calculate the above value efficiently, we notice that above summation can be regarded as propagation of a tangent vector.

We first change the network definiton into

$$\begin{aligned}
x_0(\theta, s_a) &= s_a, \\
y_i(\theta, s_a) &= c_i W_i x_{i-1}(\theta, s_a), \quad \forall i = 1, \ldots, d, \\
x_i(\theta, s_a) &= \sigma_i(y_i(\theta, s_a)) + b_i, \quad \forall i = 1, \ldots, d-1.
\end{aligned}$$

Notice that compared to original network, we add vectors $b_i$, and when $b_i$ is set as $0$, the output is same as original network.

It can be easily seen that $\nabla_{x_j} y_d = \nabla_{b_j} y_d$, therefore we could just let $\sigma_j''(y_j(\theta(t), s_a)) \left( \sum_{i=1}^{j} (\Xi^{(i,j)})_a \right)$ be the tangent vector for $b_i$ and let it propagates along the forward pass, the result tangent vector in output space is just an estimation of $\text{Tr}(H_a(\theta(t)))$.

### F.4 Final PAC Bayes bound

$$\begin{aligned}
\mathbb{E}_{\theta \sim q_t}[R(\theta)] &= \Phi_{\frac{\lambda}{m}}^{-1} \left( \mathbb{E}_{\theta \sim q_t}[r(\theta)] + \frac{D_{KL}(q_t||p) + \log \frac{1}{\delta}}{\lambda} \right) \\
D_{KL}(q_t||p) &= \mathbb{E}_{\{(Y_0^{(i)}, \Gamma_0^{(i)})\}_{i=1}^d \sim \Sigma}[V_t - S_t] \\
V_0 &= 0, S_0 = 0 \\
Y_t &= Y_t^{(d)} \\
\frac{d}{dt} V_t &= -\nabla \mathcal{L}(Y_t) \sum_{i=1}^d Y_t^{(i)} \\
\frac{d}{dt} S_t &= -Tr(\nabla^2 \mathcal{L}(Y_t) \Theta^{(d)}) - \nabla \mathcal{L}(Y_t) \sum_{j=1}^d \sum_{i=1}^j \Gamma^{(i)} \odot \Xi^{(i,j)} \\
\frac{d}{dt} Y_t^{(i)} &= -\Theta^{(i)} \nabla \mathcal{L}(Y_t)^\top \\
\frac{d}{dt} \Gamma_t^{(i)} &= -\Phi^{(i)} \nabla \mathcal{L}(Y_t)^\top
\end{aligned}$$

$$(55)$$

The closed form solution of the KL divergence is hard to obtain, but the numerical solution can be computed efficiently on a computer (as shown in Figure 4) by solving the above ODE.

