# OpenReview forum: "Optimization and Bayes: A Trade-off for Overparameterized Neural Networks"
_NeurIPS.cc/2023/Conference — NeurIPS 2023 poster_

### Official Review · Reviewer_4jd4 · 2023-07-04

**Soundness:** 2 fair
**Presentation:** 1 poor
**Contribution:** 1 poor
**Rating:** 4
**Confidence:** 3

**Summary:**

The paper presents a novel approach that bridges the gap between optimization and Bayesian learning problems. It introduces an interpolation technique that connects optimization and Bayesian learning. Additionally, the author proposes a generalization bound for infinite wide neural networks and establishes a connection between generalization and sampling efficiency.

**Strengths:**

The primary strength of this work lies in its successful formulation of a generalization bound for infinite wide neural networks. The author also provides insightful insights into the relationship between generalization and sampling efficiency, shedding light on an important aspect of the problem.

**Weaknesses:**

**Technical comments:**

- It would be beneficial to elucidate the motivation behind the sampling efficiency, denoted as $\text{eff}_{\lambda}$ in Equation (3).
- In the related works section, the author should consider mentioning alternative approaches to over-parametrized neural networks, such as random features and mean-field regimes, as suggested in [1].
- Corollary 6 is based on a surrogate loss function. Are there any assumptions on the main loss function necessary to derive an upper bound on expected loss?

**Presentation comments:**

- In the problem setup section (Line 88), the input and label spaces are denoted as $\mathcal{X}$ and $\mathcal{Y}$, respectively. However, in Line 90, the training dataset is denoted as $(s,t)$. For consistency, it would be advisable to use $(x,y)$ to represent each data sample throughout the manuscript.

- In the equation following Line 127, the right-hand side of the second equality represents an expectation with respect to $\theta \sim p(f(\theta))$. However, the current notation is unclear and could be misinterpreted as $\theta \sim p(\theta)$. This should be clarified.

- It is recommended to provide a reference for Donsker and Varadhan, which is used to derive Equation (4).

- Please provide definitions for effective sample size and sample size before Equation (3).

- Please provide a clear definition of generalization and generalization bound within the context of this paper. Is the bound on true risk?

- The theorems should be self-contained. In Line 295, the author mentions $\alpha$ without prior explanation in Theorem 5.

**Minor**

- The author employs $t_a$ and $t$ to represent label and time, respectively. It would be preferable to use alternative symbols to avoid confusion.

- Inline equations in Lines 637 and 641 should be displayed as outlined equations for improved readability.

**References:**

-       [1]: Fang, Cong, Hanze Dong, and Tong Zhang. "Mathematical models of overparameterized neural networks." Proceedings of the IEEE 109.5 (2021): 683-703.




**Questions:**

- Please elaborate more on the sentence in lines 178-179: “This is also the gap between expected loss bound obtained by some training methods and the optimal one.”
- For binary classification, there are different surrogate loss functions. Please provide some examples of surrogate loss function in multi-classification.


**Limitations:**

-    The concept of importance sampling is not new, and the author proposes importance sampling algorithms to solve Bayesian learning problems. However, the paper does not explicitly discuss the variance issues associated with importance sampling, which can be observed in the experiment results (Figure 3, left).

-    The analysis for over-parametrized neural networks may not be applicable to networks with activation functions like ReLU or other functions with unbounded first derivatives.

---

> ### Author Rebuttal · Authors · 2023-08-10
>
> Thank you for your detailed feedback. We appreciate the time and effort you've spent in reviewing our work. Below, we've addressed your points in the order presented.
>
> > It would be beneficial to elucidate the motivation behind the sampling efficiency, denoted as $\text{eff}_{\lambda}$ in Equation (3).
>
> Thanks for the clear question. We are willing to explain sampling efficiency in more detail.
>
> We've defined the sampling efficiency as $(\mathbb{E}[w])^2/\mathbb{E}[w^2]$, which is the ratio of effective sample size $n_e^*=\frac{n(\mathbb{E} [w])^2}{\mathbb{E} [w^2]}$ to the actual sample size $n$. The effective sample size is a key concept in importance sampling, controlling the variance of the final estimator.
>
> To provide a brief overview of the concept, we now provide an elementary derivation of the form of sampling efficiency and effective sample size in the importance sampling context. In importance sampling, $w$ represents the importance weights. Given $n$ samples, each of which is independently sampled as $Z_i$, and the importance weights are $w_i$. The final estimate is $S=\frac{\sum_{i=1}^n w_i Z_i}{\sum_{i=1}^n w_i}$. If the variance of $Z_i$ itself is $\sigma^2$, the variance of $S$ is $\frac{\sum_{i=1}^n w_i^2 \sigma^2}{(\sum_{i=1}^n w_i)^2}$. We know that the variance of the mean of $n_e$ number of independent identically distributed variables is $\frac{\sigma^2}{n_e}$. Thus, the final variance in importance sampling is **equivalent** to the variance when sampling $n_e=\frac{(\sum_{i=1}^n w_i)^2}{(\sum_{i=1}^n w_i^2}$ independent samples. Therefore, the effective sample size is defined as $n_e=\frac{(\sum_{i=1}^n w_i)^2}{(\sum_{i=1}^n w_i^2}=\frac{n\overline{w}^2}{\overline{w^2}}$. Since $w_i$ is also a random variable, we can derive the population version of $n_e$ by changing the sum to the expectation. Thus, we have $n_e^*=\frac{n(\mathbb{E} [w])^2}{\mathbb{E} [w^2]}$. The effective sample size is generally less than the true sample size, and this decrease factor is the sampling efficiency, which leads to the definition in Eq. (3).
>
> We hope this clarifies your question. More details can be found in Owen, Art B. "Monte Carlo theory, methods and examples." (2013). (Section 9.3)
>
> > the author should consider mentioning alternative approaches to over-parametrized neural networks, such as random features and mean-field regimes, as suggested in [1].
>
> Thank you for the suggestion. We will add references to these alternative approaches in the related works section.
>
> > Corollary 6 is based on a surrogate loss function. Are there any assumptions on the main loss function necessary to derive an upper bound on expected loss?
>
> The only requirement we have for the main loss function is that its output is in $[0,1]$, as stated in line 91.
>
> > For consistency, it would be advisable to use $(x,y)$ to represent each data sample throughout the manuscript.
>
> Thank you for your suggestion. We understand the value of consistent notation. However, in Section 7, 'x' and 'y' are used to represent activations in neural networks. To prevent confusion, we opted to use 's' and 't' to denote inputs and outputs in the manuscript.
>
> > In the equation following Line 127, the right-hand side of the second equality represents an expectation with respect to $\theta \sim p(f(\theta))$. However, the current notation is unclear and could be misinterpreted as $\theta \sim p(\theta)$. This should be clarified.
>
> It appears that your understanding of this equation is incorrect.
>
> We confirm that we intended to write $\theta \sim p(\theta)$, not $\theta \sim p(f(\theta))$. In fact, the latter notation would not make sense as $p(f(\theta))$ cannot be regarded as a distribution. If you could explain why you believe it should be $\theta \sim p(f(\theta))$, perhaps we can better assist in clarifying this misunderstanding.
>
> > It is recommended to provide a reference for Donsker and Varadhan, which is used to derive Equation (4).
>
> We'll add the following citation:
> "Donsker, Monroe D., and SR Srinivasa Varadhan. "Asymptotic evaluation of certain Markov process expectations for large time—III." Communications on pure and applied Mathematics 29.4 (1976): 389-461."
>
> > Please provide definitions for effective sample size and sample size before Equation (3).
>
> We will add definition of effective sample size in the manuscript. For more details about effective sample size , olease kindly refer to our explanation in the beginning of the response.
>
> > Please provide a clear definition of generalization and generalization bound within the context of this paper. Is the bound on true risk?
>
> Generalization refers to the phenomenon that minimizing the empirical loss on a training dataset leads to reduced  expected loss on new, unseen data. Generalization bound is just an upper bound of the difference between the expected loss and the empirical loss. For a precise definition, please refer to lines 88-97.
>
> We use $R(h)$ to denote true risk, specifically, the expected prediction error of a model on the unknown real data distribution. So, yes. The upper bound is on true risk.
>
> > In Line 295, the author mentions $\alpha$ without prior explanation in Theorem 5.
>
> We agree with your observation. We will clarify this by stating "for all $\alpha=1,\dots,j$" in the Theorem.
>
> > It would be preferable to use alternative symbols to avoid confusion.
>
> Thank you for your suggestion. We will change symbol $t$ to $z$ for notation of label.
>
> > Inline equations in Lines 637 and 641 should be displayed as outlined equations for improved readability.
>
> Thank you for pointing this out. We will make the necessary adjustments to improve readability.

---

> > ### Author Response · Authors · 2023-08-10
> > **Additional Response by Authors**
> >
> > > Please elaborate more on the sentence in lines 178-179: “This is also the gap between expected loss bound obtained by some training methods and the optimal one.”
> >
> > That sentence refers to the second term on the right side of Donsker and Varadhan’s variational formula in equation (4). Donsker and Varadhan’s variational formula, proposed in the last century, plays a significant role in PAC Bayesian, and its proof is quite simple – one can derive it by unfolding the definition.
> >
> > The second term on the right side of Donsker and Varadhan’s variational formula measures the difference between the distribution $q$ and the Gibbs distribution $p_\lambda$. Only this second term on the right side depends on $q$, while the first term does not.
> >
> > The left side of the equation represents the part related to $q$ in the PAC Bayesian bound. We aim to optimize this equation to obtain the best bound for the expected loss. According to Donsker and Varadhan’s variational formula, if $q$ is the optimal Gibbs distribution $p_\lambda$, the whole equation is minimized. If not, a sub-optimal bound would result, and the gap between this sub-optimal bound and the optimal bound can be measured by a KL divergence.
> >
> > We hope this clarifies the concept for you.
> >
> > > For binary classification, there are different surrogate loss functions. Please provide some examples of surrogate loss function in multi-classification.
> >
> > Certainly, here are several examples of surrogate loss functions that are commonly used in multi-classification tasks:
> > 1. Logistic Loss:
> > $$L(y, z) = -\sum_{i=1}^{C} z_i \log \left( \frac{e^{y_i}}{\sum_{j=1}^{C} e^{y_j}} \right)$$
> > 3. Hinge Loss:
> > $$L(y, z) = \sum_{j \neq z} \max(0, y_j - y_z + \Delta)$$
> > 3. Squared Loss:
> > $$L(y, z) = \sum_{i=1}^{C} (y_i - z_i)^2$$
> > 4. Exponential Loss:
> > $$L(y, z) = \sum_{i=1}^{C} \exp(-y_i z_i)$$
> > We hope that answers your question.
> >
> > > The concept of importance sampling is not new, and the author proposes importance sampling algorithms to solve Bayesian learning problems. However, the paper does not explicitly discuss the variance issues associated with importance sampling, which can be observed in the experiment results (Figure 3, left).
> >
> > First, we concur that importance sampling is a classic method, and our method is a class of importance sampling algorithms.
> >
> > Second, we respectfully disagree about the claim "the paper does not explicitly discuss the variance issues". We do address the variance issue of importance sampling through the concept of effective sample size.
> >
> > The effective sample size and the variance of importance sampling are inversely related: as the effective sample size increases, the variance decreases. Indeed, the concept of effective sample size is derived from the observation of variance in importance sampling. Regarding your previous questions about effective sample size, we hope our explanation in the beginning of the response clarifies its definition and its impact on variance in importance sampling.
> >
> > > The analysis for over-parametrized neural networks may not be applicable to networks with activation functions like ReLU or other functions with unbounded first derivatives.
> >
> > We believe our paper still provides valuable insights, even though the analysis may not directly transfer to activation functions like ReLU.
> >
> > Lastly, we appreciate your detailed comments and questions. We hope our reply has addressed your concerns and provided clarifying context where it was needed. Please feel free to ask if you have further questions or need additional information.

---

> > > ### Comment · Reviewer_4jd4 · 2023-08-14
> > > **Response to rebuttal**
> > >
> > > I appreciate the author for their response. Most of my concerns are answered.
> > >
> > > Due to the following concerns
> > > - There is no comparison with other methods which apply to over-parameterisation regime,
> > > - The application of the method is limited for bounded activation function,
> > > - The method is based on importance sampling which the variance problem is not discussed explicitly,
> > >
> > >
> > > and more important that these limitations are not discussed in the paper, therefore I slightly increase my evaluation score to 4.

---

### Official Review · Reviewer_TNvC · 2023-07-05

**Soundness:** 4 excellent
**Presentation:** 4 excellent
**Contribution:** 3 good
**Rating:** 7
**Confidence:** 3

**Summary:**

The paper provides a perspective interpolating optimization and Bayesian inference. The gradient based optimization procedure gives an output distribution $q(x)$ and the Bayesian inference procedure gives a posterior $p_\lambda(x)$, the two distributions are connect by the weight $w_\lambda$. Then the paper proposes to find an interpolation of gradient based optimization and Bayesian inference by using a different weight $w_\lambda^\beta$.

**Strengths:**

1. The paper is very well written, even for a reader with not much background in optimization, I found the paper very easy to follow and the main results clearly presented.
2. Although I am not familiar with the literature of optimization, but I feel the interpolation of gradient based optimization and bayesian inference is an important contribution to the community.
3. The paper proposes the entropy term and the energy term as two major terms governing the weight, and hence measures the sampling efficiency and generalization error bound. For finite fully connected layers, the analytic form of energy and entropy is provided, and also empirically studied. The paper provides a promising perspective for future analysis of other types of deep neural networks like CNNs and transformers.


**Weaknesses:**

Only some minor ones.

Minor weakness:
1. line 275, equation 9 is too long
2. line 265, Equation (27) is not properly referenced.


**Questions:**

1. Some of the main theoretical result still follows from the basic assumption of infinite width limit of fully connected neural networks, which is a commonly used approach in theoretical analysis of deep neural networks. However, it is well known that in practice this assumption is generally not true, so I am curious to know what can be improved to generalize the theoretical setting to finite width neural networks.

2. Why the paper considers specifically a simple clipping weight, rather than the actual power function $w_\lambda^\beta$, does the proof require that specific form of weight to hold true?


**Limitations:**

Yes.

---

> ### Author Rebuttal · Authors · 2023-08-10
>
> We sincerely appreciate your constructive feedback and recognition of our work. Your comments have helped us to improve our manuscript.
>
> > Weaknesses
>
> Thank you for the suggestion. We will improve the presentation of the mentioned equations for clarity.
>
> > Some of the main theoretical result still follows from the basic assumption of infinite width limit of fully connected neural networks, which is a commonly used approach in theoretical analysis of deep neural networks. However, it is well known that in practice this assumption is generally not true, so I am curious to know what can be improved to generalize the theoretical setting to finite width neural networks.
>
> Indeed, this is a critical question. The assumption of infinite width made the theoretical analysis in Section 7 feasible. However, such assumption is not always valid, and this is a common challenge in many works analyzing infinitely wide networks. To address this, some researchers [1,2,3] have started to explore the extension of infinite width network analysis to finite widths. We believe that an important direction for future work would be to examine whether similar techniques can be extended to our setup, to provide estimation error for the entropy term and energy term under finite width networks.
>
> > Why the paper considers specifically a simple clipping weight, rather than the actual power function $w_\lambda^\beta$, does the proof require that specific form of weight to hold true?
>
> This is a great insight. In our paper, we considered weight clipping due to the monotonicity guarantee provided in Theorem 3. However, this is not the only possible choice. Following your recommendation, we found that the power function $v_\beta(w_\lambda) = w_\lambda^\beta$ for $\beta \in [0,1]$ can also creates an interpolation between optimization and Bayesian learning. When $\beta = 0$, the modified weights are constant at 1, which turns the distribution into a deep ensemble. When $\beta = 1$, the weights remain unchanged, preserving the Gibbs distribution. This power function also maintains a monotonicity property similar to that in Theorem 3. Therefore, it is indeed feasible, and we appreciate your suggestion.
>
> We express our gratitude once again for your thoughtful comments and questions. Your insightful suggestions have inspired us to explore further.
>
> [1] Hanin, Boris, and Mihai Nica. "Finite depth and width corrections to the neural tangent kernel." arXiv preprint arXiv:1909.05989 (2019).
>
> [2] Littwin, Etai, Tomer Galanti, and Lior Wolf. "On random kernels of residual architectures." Uncertainty in Artificial Intelligence. PMLR, 2021.
>
> [3] Dyer, Ethan, and Guy Gur-Ari. "Asymptotics of wide networks from feynman diagrams." arXiv preprint arXiv:1909.11304 (2019).

---

> > ### Comment · Reviewer_TNvC · 2023-08-15
> >
> > Thank you for your response.
> >
> > After reading the author's rebuttal and other fellow reviewers' comments, I do not see any explicit reason for me to change my score and hence I still recommend acceptance.

---

### Official Review · Reviewer_5yu1 · 2023-07-06

**Soundness:** 2 fair
**Presentation:** 2 fair
**Contribution:** 3 good
**Rating:** 6
**Confidence:** 3

**Summary:**

This paper proposes a framework to bridge the gap between ERM and Bayesian learning problems.
The authors derived the algorithm-dependent PAC-Bayesian generalization bound for infinitely wide networks where the KL divergence between the posterior distribution obtained by infinitesimal step size gradient descent and a Gaussian prior.
Since direct theoretical analysis on the KL divergence is intractable, they used the well-used simplification that the output distribution of  neural networks drawn from the prior can be approximated by a Gaussian distribution in the infinite width limit.
From the analysis, they provided an interpolation method for accuracy-computation trade-off.
In addtion, as a byproduct, this paper analyze the dynamics of the Hessian trace.

**Strengths:**

This paper presents new insights into the relationship between ERM and Bayesian learning.
The most interesting aspects of this paper are the analysis of the KL divergence using variable transformations and its relation to the change in Helmholtz free energy in isothermal processes.
Moreover, the dynamics of the Hessian trace, which is typically relate to flatness in loss landscape, is analyzed.


**Weaknesses:**

Although the motivation is interesting and the derivations of the equations are interesting, it is difficult to understand what novel PacBayes bound was finally obtained.
It is also difficult to understand how they differ from the previous PacBayes bound and what kind of findings are obtained.
Many of these problems seem to be presentation issues rather than technical issues.


**Questions:**

Q.1 Would you write explicitly what the final resulting of the novel PAC Bayes bound is obtained?
Since most of the analysis is in the dynamics of the KL divergence, the paper is gradually being analyzed for KL.
However,  makes it difficult to understand the final results obtained.
In particular, I would like to know the final result of the PAC Bayes bound formulated as a function of training time and training data.

Q.2 I would like to know if it is possible to compare the results of Gaussian processes with those in the limit of infinitely wide neural networks.
A neural network with infinite width can be regarded as a Gaussian process, and the PacBayes bound of the Gaussian process can be obtained by Seeger by using the Radon-Nidodim derivative.
Very naively , it seems that the PacBayes bound can be obtained by plugging the NTK derived from the infinite width of neural network into the PacBayes bound of the Gaussian process.
If this Gaussian process based analysis can be an answer of Q1 in the paper, I would like to know what differences can be observed compared to this case.


**Limitations:**

The structure of the paper is difficult to understand.
The theoriy that are ultimately obtained is unclear compared to the exisitng PAC Bayes bound.

---

> ### Author Rebuttal · Authors · 2023-08-10
>
> Thank you for taking the time to review our work and for your insightful comments and questions. We appreciate your recognition of the novel insights and contributions of our work in bridging the gap between ERM and Bayesian learning.
>
> > Would you write explicitly what the final resulting of the novel PAC Bayes bound is obtained? ...  I would like to know the final result of the PAC Bayes bound formulated as a function of training time and training data.
>
> The final result of the PAC Bayes bound includes several parts, a main bound and a decomposition of KL divergence, and  dynamics of energy and entropy change. The main PAC-Bayes bound is:
>
> $\mathbb{E}_{\theta\sim q_t}[R(\theta)]=\Phi^{-1}\_{\frac{\lambda}{m}}(\mathbb{E}\_{\theta\sim q\_t}[r(\theta)]+\frac{D\_{KL}(q\_t||p)+\log\frac{1}{\delta}}{\lambda})$
>
> where the KL divergence $D_{KL}(q_t||p)$ is given by:
>
> $D\_{KL}(q\_t||p)=\mathbb{E}\_{\\{(Y^{(i)}\_0,\Gamma^{(i)}\_0)\\}_{i=1}^d\sim\Sigma} [V\_t-S\_t] $
>
> The dynamics of $V_t$ and $S_t$ are given by the differential equations:
>
> $\begin{align*}
> V\_0&=0,S\_0=0\\\\
> \frac{d}{dt}V_t &= -\nabla \mathcal{L}(Y_t)\sum_{i=1}^d Y^{(i)}_t\\\\
> \frac{d}{dt}S_t &= -\text{Tr}(\nabla^2\mathcal{L}(Y_t)\Theta^{(d)})-\nabla\mathcal{L}(Y_t)\sum\_{j=1}^d\sum\_{i=1}^j \Gamma^{(i)}\odot\Xi^{(i,j)}\\\\
> \frac{d}{dt}Y_t&=-\Theta^{(i)}\nabla \mathcal{L}(Y_t)^\top\\\\
> \frac{d}{dt}\Gamma_t&=-\Phi^{(i)}\nabla \mathcal{L}(Y_t)^\top
> \end{align*}$
>
> The closed form solution of the KL divergence is hard to obtain, but the numerical solution can be computed efficiently on a computer (as shown in Fig 4 of our paper) by solving the above ODE.
>
> > If this Gaussian process based analysis can be an answer of Q1 in the paper, I would like to know what differences can be observed compared to this case.
>
> Thank you for this thought-provoking question.
>
> You're correct in stating that a neural network with infinite width can be regarded as a Gaussian process, but this relationship only holds at initialization. This correspondence breaks down once the network is trained via gradient descent, leading to a distribution that differs from a Gaussian process. While one can indeed construct a Gaussian process using the conjugate kernel $\Sigma$ (as opposed to the NTK $\Theta$) to answer questions about the Bayesian learning of neural network, it cannot directly answer questions about the generalization of neural networks trained by gradient descent, hence it cannot be used to answer the Q1.
>
> Thank you once again for your valuable feedback and insightful questions. We hope  this response have addressed your concerns and questions.

---

### Official Review · Reviewer_sSya · 2023-07-06

**Soundness:** 3 good
**Presentation:** 2 fair
**Contribution:** 3 good
**Rating:** 5
**Confidence:** 2

**Summary:**

The paper introduces a novel learning algorithm, Transformative Bayesian Learning (TransBL), which aims to navigate the balance between empirical risk minimization (ERM) and Bayesian learning in the context of overparameterized neural networks. The authors establish a PAC-Bayesian bound for these networks and provide a theoretical discourse on the relationship between generalization and sampling efficiency. Their primary goal is to explore an intermediary distribution that bridges the gap between ERM and Bayesian learning.The paper presents a sound theoretical argument which I found to be interesting and with potentially moderate impact.

The main issue currently I see is the presentation of their results. For example, the paper currently lacks certain buildup and intuitive explanations and some clarifications before giving formal treatment. Besides that, the order and flow of the paper currently uses a lot of ambiguities  until they are cleared up much later. I've added several of these ambiguities. Perhaps the authors assumed the reader to be very intimately familiar with this exact research question who will "mentally fill in the gaps", but I don't think this should be the default assumption.  I've faced some these ambiguities as I read the manuscript, and added them in the "weaknesses" section. Due to my lack of expertise on some related literature, I defer the judgement about novelty of these ideas to other reviewers.

**Strengths:**

- Sound theoretical analysis: The paper provides a theoretical analysis on the relationship between ERM and Bayesian learning, specifically in the context of overparameterized neural networks, that may be novel.

- The introduction and justification of TransBL is innovative and backed by comprehensive derivations. The concept of transforming gradient-based optimization into importance sampling could also be interesting.

- Interpolation Mechanism: The introduction of an interpolation mechanism by modifying weights presents an innovative solution to the trade-off between computation efficiency and generalization error.

- Usage of PAC-Bayesian Bounds: The derivation of algorithm-dependent generalization bounds through the use of PAC-Bayesian theory and infinitely wide neural networks adds depth to the theoretical contribution.

- Analysis of Energy and Entropy Change: The authors provide a comprehensive understanding of the dynamics of energy and entropy changes, which play a pivotal role in their algorithm. This exploration of these dynamics could pave the way for further developments in deep Bayesian learning and PAC-Bayesian bounds.

**Weaknesses:**



- While the presented theoretical results are relatively well stated, there is a lack of high level intuition and build up that make the paper rather hard to understand for anyone who's not expert on all the related topics.  This can be alleviated by using one or few relevant examples (toy examples where everyone can grasp and convey some key points, as well as by giving some more intuitive and high level explanation of the ideas, before giving a formal description.
-  It would be helpful for the authors to make some parts of the paper notations and variables clarified so that a reader can follow the terms without the need to know the cited literature or read the appendix.  I've added several questions in the "Questions" section to raise some of these points. As another example, in section 7.3 there seems to be a clarification of what $f$ entails by the equation $\theta(t)= f(t,\theta(0))$ . Some earlier mention/clarification on this would have helped me/possibly other readers. In line with previous comment, more concrete explanations, as opposed to merely abstract terms, will help clarify the ideas.

- The assumptions such as infinitely wide networks could limit the applicability of the results. Discussion on these assumptions and how they might affect the applicability of their method to practical settings with finite width will be helpful to the reader. While it is understandable that the limit of infinite width is necessary for the theoretical derivations is necessary, perhaps the authors can add more discussions on the approximation errors that this implies for practice , ie. the discrepancy between infinite for theory and finite width used in practice.

- Lack of sufficient empirical validations and baselines: While the paper can be viewed as a mere theoretical contribution, the central proposition of the paper, that they bridge the gap between ERM and Bayesian learning,  is a claim of empirical nature. In other words, more empirical evidence can substantially strengthen the validity of the work and broaden its applicability. While the authors do compare their method to ERM and Bayesian learning, there is a distinct lack of comparison with other more recent methods aiming to achieve similar objectives. These comparisons will be helpful in informing the reader about the practical implications of these study and broaden the paper's impact.  If the authors do not believe further experiments are necessary, perhaps they can be more upfront about their contribution to be of a theoretical nature.

**Questions:**


- in section 3 the term output distribution $q$ and optimisation flow $f:\Theta\to\Theta$ are introduced but are left somewhat loosely defined. My impression from reading section 3 multiple times is the following: $p(\theta)$ is the prior, namely just the weight and bias distributions of the model. The flow $f:\Theta\to\Theta$ corresponds to a single step of the gradient-based optimisation procedure (based on its domain & range sets, and specially its name), such that $f(\theta)$ corresponds to the next step of the optimisation procedure. And $q(\theta)$ is the probability of the parameters after one step of the optimization process. This interpretation make an intuitive sense: if we assume the gradient updates $\theta_{t+1} = \theta_t + \epsilon \nabla f(\theta)$ for some infinitesimal $\epsilon$ then the fact that volume of $|\nabla f(\theta)|$ appearing as a multiplicative term $q(\theta)$. I understand that I may be incorrect in these interpretation, since an alternative interpretation is that $q$ and $f$ refer to the "final" distribution over parameters (i.e., many steps of the gradient descent). Can authors clarify these points?
- The definitions given between lines 129-130 given for energy and entropy seem rather interesting but somewhat ambiguous. My question is two folds: 1)  it would be good to be clear about whether these are the contributions of this paper? It would be helpful for readers to know which parts are and are not the contributions of this work. 2) What is the intuition behind calling the log-determinant term "entropy"? It would be helpful if authors could provide a concrete example for explaining these concepts to familiarize the readers.
- In line 155 the authors proclaim that the equation (2) doesn't "involve any more training.
- Later in the text after line 261 the authors compute $d \theta(t)/d t$ and refer to it as "gradient flow". Is this gradient flow the same as "f" introduced earlier in section 3?


**Limitations:**

yes

---

> ### Author Rebuttal · Authors · 2023-08-10
>
> Thank you for the detailed and constructive feedback.
> > Strengths
>
> We are grateful for your recognition of the strengths of our paper, especially our theoretical contributions in areas including the interpolation mechanism, usage of PAC-Bayesian bounds, and analysis of energy and entropy change.
> > While the presented theoretical results are relatively well stated, there is a lack of high level intuition and build up that make the paper rather hard to understand for anyone who's not expert on all the related topics. This can be alleviated by using one or few relevant examples (toy examples where everyone can grasp and convey some key points, as well as by giving some more intuitive and high level explanation of the ideas, before giving a formal description.
>
> Thank you for this valuable suggestion about providing intuitive explanations alongside our theoretical findings. We plan to include a motivating example, which contains a single-variable loss function with two global minima, showing how our TransBL approach intuitively assigning a smaller weights to the sharp minima due to entropy change. This example will be further discussed with references to PAC Bayesian methods to give readers a clearer understanding.
> > As another example, in section 7.3 there seems to be a clarification of what $f$ entails by the equation  $\theta(t)= f(t,\theta(0))$. Some earlier mention/clarification on this would have helped me/possibly other readers.
>
> Thank you for pointing this out. We will ensure that the function $f$ is introduced and explained earlier in the manuscript.
>
> > While it is understandable that the limit of infinite width is necessary for the theoretical derivations is necessary, perhaps the authors can add more discussions on the approximation errors that this implies for practice , ie. the discrepancy between infinite for theory and finite width used in practice.
>
> We concur that infinite width is a strong assumption. Such assumption made the theoretical analysis in Section 7 feasible, and it is prevalent in many theoretical discussions, including the prior work mentioned in lines 43-44. There are some works attempting to analyze finite-width corrections. However, these often complicate the theoretical analysis, and we hope that reviewer could understand our choice to leave such discussions for future work. Nevertheless, we appreciate your suggestion and will enhance our discussion on the necessity and limitations of the infinite width assumption.
>
> > Lack of sufficient empirical validations and baselines
>
> Thank you for raising this important point. Our main contribution is indeed theoretical, bridging the ERM and Bayesian learning through a parameter $\beta$. This paradigm is a departure from previous works, making direct comparisons challenging in some aspects.
>
> Existing variational approximation methods can also be viewed as a compromise between ERM and Bayesian learning. However, our method significantly differs from such variational approximation. As demonstrated in our experiments, especially as shown in Fig 3.a, we achieve continuous interpolation. On the other hand, variational approximation's bias depends on the expressiveness of the function class, which means theoretical guarantee of connecting ERM and Bayesian learning and achieving a continuous control like in Fig 3.a are not possible. This makes direct comparisons with these variational approximation difficult.
>
> Apart from our main theoretical contribution, we offer a practical contribution. We've developed a novel method for estimating entropy change, as detailed in Sec 8.1. Our method proves to be notably more efficient than prior approaches based on the Hutchinson method.
>
> > And $q(\theta)$ is the probability of the parameters after one step of the optimization process
>
> > alternative interpretation is that $q$ and $f$ refer to the "final" distribution over parameters (i.e., many steps of the gradient descent). Can authors clarify these points?
>
> You are correct in the explanation of $q(\theta)$. Both "one step" and "many steps" interpretations are valid and do not affect the analyses within sections 3-6. "f" can represent any invertible function with an existing Jacobian determinant, encompassing single-step gradient descent (GD), multi-step GD, and gradient flow. Only in Section 7 do we focus on gradient flow to discuss the specific forms of energy and entropy change.
>
> > it would be good to be clear about whether these are the contributions of this paper? It would be helpful for readers to know which parts are and are not the contributions of this work
>
> The energy and entropy decomposition of KL divergence in lines 129-130 is based on a straightforward analogy. We don't regard this definition itself as a contribution of this work. Our four main contributions are detailed from lines 68-84.
>
> > What is the intuition behind calling the log-determinant term "entropy"? It would be helpful if authors could provide a concrete example for explaining these concepts to familiarize the readers.
>
> Consider a one-dimensional function f(x)=2x. The Jacobian here is 2. If input x is uniformly distributed within [0,1], the output f(x) will be uniformly distributed within [0,2]. The entropy increases by ln(2), which corresponds to the log-determinant. We'll include this example in our revisions to make our definitions more intuitive.
>
> > In line 155 the authors proclaim that the equation (2) doesn't "involve any more training.
>
> Yes, equation (2) doesn't involve any additional training of the parameter $\theta$ beyond the optimization procedure in optimization flow f.
>
> > Is this gradient flow the same as "f" introduced earlier in section 3?
>
> Yes, the gradient flow discussed in line 261 is the same as the abstract "f" discussed in sections 3-6. The analyses in Sections 3-6 deal with "f" in an abstract, general sense. Only in Section 7 do we narrow our analysis to the gradient flow.
>
> Thank you again for your time and insightful comments.

---

> > ### Comment · Reviewer_sSya · 2023-08-21
> >
> > I thank the authors for their clarifications and taking the time to answer my questions.
> > Since most of the points raised were with regard to presentation, I hope that authors will take care of them if the paper is accepted.
> > While I do understand the reasoning of authors for studying the infinite width settings, merely tending width to infinity can lead to highly simplified dynamics (kernel gradient descent) that authors rely on to derive their results, which severely limits how predictive and interesting these results are for real settings. For this reason I will keep my score as borderline accept.

---

### Official Review · Reviewer_1HBf · 2023-07-08

**Soundness:** 3 good
**Presentation:** 2 fair
**Contribution:** 3 good
**Rating:** 5
**Confidence:** 2

**Summary:**

The paper derives an algorithm-dependent PAC-Bayesian generalization bound for infinitely wide neural networks that is based on the KL divergence between posterior obtained by gradient flow and a Gaussian prior. The work shows how to transform optimization into importance sampling through interpolation, coined TransBL, to trade-off generalization and sample efficiency. Also, it provides a proof of non-diminishing Hessian trace under certain conditions.

**Strengths:**

- The paper is clearly structured and the questions it aims to answer are well motivated.



- Interesting function space perspective of neural network training dynamics through analysis of the KL divergence under infinite width limit assumptions.
- Bridges gap between optimization and Bayesian learning problems to allow interpolation between computational efficiency and predictive accuracy.
- Additional results of Hessian trace that could be independently useful.

**Weaknesses:**

- Experimental setup sometimes hard to follow

It was sometimes hard to follow the setup of the experimental section (maybe dedicate a separate section in the appendices for details). Having a better picture of the details (especially how inference for TransBL is performed) would allow reproducibility but also spot details that could cause misalignment between theory and empirical results.

- Limited experiments

The results are promising and already clearly show that the theory is already well-aligned with learning in practice. But, it would be interesting to extend this to a slightly wider set of problems or configurations.

**Questions:**

a) There seems to be a slight drop in accuracy for TransBL in Figure 3.a. Is it expected that this is due to noise?

b) The work seems to rely on gradient-flow with infinitesimal step size, whereas regular neural networks are typically trained with finite step sizes. To what extent can we expect similar results still?

c) What happens empirically for different $\beta$?



**Limitations:**

- experimental setup could be more clear

The experimental set-up is not very clear to me. This makes it harder to judge how well the empirical results align with the theoretical derivations.

- discussion on current limitations

The paper compares learning in practice with theoretical results. It would be helpful if the paper could also assess in what cases we would expect current results not to hold anymore.

- typos

In the abstract, the method is called 'TansBL' (twice), whereas the paper uses 'TransBL'.

---

> ### Author Rebuttal · Authors · 2023-08-10
>
> We deeply appreciate your recognition of the clarity, motivation, and contributions of our work.
>
> > It was sometimes hard to follow the setup of the experimental section (maybe dedicate a separate section in the appendices for details). Having a better picture of the details (especially how inference for TransBL is performed) would allow reproducibility but also spot details that could cause misalignment between theory and empirical results.
>
> Thank you for the suggestion. We will incorporate a more detailed description of the experimental setup in the appendices, providing specifics like the ensemble sampling used and an explanation for the shading in Fig 3.a.
>
> To address reproducibility concerns, we've provided the code with seeds in the supplementary material to ensure consistent replication of our results.
>
> > it would be interesting to extend this to a slightly wider set of problems or configurations
>
> We appreciate your recommendation. One of the main challenges in expanding the experiments is our focus on the distribution obtained from optimization, rather than sampling few solutions as done in many deep learning studies. For instance, to produce Fig 3.b, we ran over 100,000 optimization processes. This complexity makes it challenging to quickly expand to other configurations. However, we do recognize the importance and potential of broader applicability, and we will explore this in future work.
>
> > There seems to be a slight drop in accuracy for TransBL in Figure 3.a. Is it expected that this is due to noise?
>
> We believe this is likely due to statistical noise. As indicated by the broad shading (which represents a 3σ interval), the variance is quite high in the early training stages. At this phase, due to the initial parameter distribution not aligning with the Gibbs distribution, the effective sample size is relatively small, resulting in a larger variance. Consequently, the estimated mean can be noisy.
>
> > The work seems to rely on gradient-flow with infinitesimal step size, whereas regular neural networks are typically trained with finite step sizes. To what extent can we expect similar results still?
>
> Thank you for raising this important point. We opted for the infinitesimal step size primarily to simplify the analysis, as it allows us to describe neural network training dynamics with an ODE. For finite step sizes, we believe the core ideas could still be applicable with discrete step sizes, although the theoretical analysis might be more challenging. Moreover, in our experiment with actual neural networks, we indeed use discrete step sizes and have observed that the empirical results closely align with our theoretical findings, as shown in Fig 2.
>
> >  What happens empirically for different $\beta$?
>
> Empirically, β serves as a control between generalization and sample efficiency. Larger values tend to exhibit behavior closer to a Bayesian posterior with a rapid decrease in effective sample size but improved generalization, as illustrated in Fig 3.c. Smaller values, on the other hand, show behavior closer to standard deep ensembles.
>
> > experimental setup could be more clear
>
> We appreciate your suggestion. Due to space limitations, we had to omit certain details from the main paper, like the aforementioned experimental parameters and figure explanations. However, in light of your feedback, we'll ensure that they're included in the appendices to enhance the clarity.
>
> > It would be helpful if the paper could also assess in what cases we would expect current results not to hold anymore.
>
> This is a pertinent point. Our current analysis heavily relies on the overparameterization. Real-world neural networks often do not operate in this regime, creating a discrepancy between the theoretical entropy estimation and reality. However, if an estimate for this can be obtained, our analysis in Sections 3-6 remains valid, and methods like TransBL and weight clipping can still be applied.
>
> > In the abstract, the method is called 'TansBL' (twice), whereas the paper uses 'TransBL'.
>
> Thank you for spotting this oversight. We apologize for the inconsistency and will correct this typo.
>
> Again, we appreciate your feedback, which will undoubtedly strengthen our manuscript.

---

### Decision · Program_Chairs · 2023-09-21

**Decision:**

Accept (poster)

**Comment:**

This meta review is based on the reviews, the authors rebuttal and the discussions with the reviewers, discussions with the SAC, and ultimately my own judgement on the paper. There was a consensus that the paper contributes sound and interesting contributions. I feel this work deserves to be featured at NeurIPS and will attract interest from the community. I would like to personally invite the authors to carefully revise their manuscript to take into account the remarks and suggestions made by reviewers. Congratulations!